# DeepDRK: Deep Dependency Regularized Knockoff for Feature Selection

## Abstract

Model-X knockoff, among various feature selection methods, received much attention recently due to its guarantee on false discovery rate (FDR) control. Subsequent to its introduction in parametric design, knockoff is advanced to handle arbitrary data distributions using deep learning-based generative modeling. However, we observed that current implementations of the deep Model-X knockoff framework exhibit limitations. Notably, the "swap property" that knockoffs necessitate frequently encounter challenges on sample level, leading to a diminished selection power. To overcome, we develop "Deep Dependency Regularized Knockoff (DeepDRK) [1]", a distribution-free deep learning method that strikes a balance between FDR and power. In DeepDRK, a generative model grounded in a transformer architecture is introduced to better achieve the "swap property". Novel efficient regularization techniques are also proposed to reach higher power. Our model outperforms other benchmarks in synthetic, semi-synthetic, and real-world data, especially when sample size is small and data distribution is complex.

## 1 Introduction

Feature selection (FS) has drawn tremendous attention over the past decades, due to rapidly increasing data dimension (Guyon & Elisseeff, 2003). Still, perfectly locating informative features is deemed mission impossible (Sudarshan et al., 2020). It is hence necessary to devise algorithms to select features with controlled error rates.

Targeting this goal, Model-X knockoffs, a novel framework, is proposed in Barber & Candès (2015); Candes et al. (2018) to select relevant features while controlling the false discovery rate (FDR). In contrast to the classical setup, where assumptions on the correlations between input features and the response are imposed (Benjamini & Hochberg, 1995; Gavrilov et al., 2009), the Model-X knockoff framework only requires a linear relationship between the response and the features. With a strong finite-sample FDR guarantee, Model-X knockoff saw broad applications in domains such as biology, neuroscience, and medicine, where the size of data is limited (Sudarshan et al., 2020).

There have been considerable developments of knockoffs since its debut. In scenarios where feature distributions are complex, various deep learning methods have been proposed. However, we observe major limitations despite improved performance. First, the performances of existing methods vary across different data distributions. Second, selection quality deteriorates when observation number is relatively small. Third, the training is often difficult, due to competing losses in the training objective. We elaborate on the drawbacks in sections 2.2 and 3.2.

In this paper, we remedy the issues by proposing the Deep Dependency Regularized Knockoff (DeepDRK), a deep learning-based pipeline that utilizes the Vision Transformer (Dosovitskiy et al., 2020) as the backbone model to generate knockoffs for feature selection. DeepDRK is designed to achieve the so-called "swap property" (Barber & Candès, 2015) and to reduce "recontructabililty" (Spector & Janson, 2022), which in turn controls FDR and boosts selection power. We propose a "multi-swapper" adversarial training procedure to enforce the swap property, while

---

[1] The "DeepDRK" term is pronounced as "Deep Dark" and is inspired by the underground cave biome prevalent in the world of Minecraft, whereas the proposed feature selection model acts as the guiding beacon amidst the darkness of unknown.

a sliced-Wasserstein-based (Li et al., 2023) dependency regularization together with a novel perturbation technique are introduced to reduce reconstructability. We then carry out experiments on real, synthetic, and semi-synthetic datasets to show that our pipeline achieves better performance in different scenarios comparing to existing ones.

## 2 BACKGROUND AND RELATED WORKS

### 2.1 MODEL-X KNOCKOFFS FOR FDR CONTROL

The Model-X knockoffs framework consists of two main components. Given the explanatory variables $X = (X_1, X_2, \ldots, X_p)^\top \in \mathbb{R}^p$ and the response variable $Y$ ($Y$ continuous for regression and categorical for classification), the framework requires: 1. a knockoff $\tilde{X} = (\tilde{X}_1, \tilde{X}_2, \ldots, \tilde{X}_p)^\top$ that "fakes" $X$; 2. the knockoff statistics $W_j$ for $j \in [p]$ that assess the importance of each feature $X_j$. The knockoff $\tilde{X}$ is required to be independent of $Y$ conditioning on $X$, and must satisfy the swap property:

$$(X, \tilde{X})_{\text{swap}(B)} \overset{d}{=} (X, \tilde{X}), \quad \forall B \subset [p]. \tag{1}$$

Here swap($B$) exchanges the positions of any variable $X_j$, $j \in B$, with its knockoff $\tilde{X}_j$. The knockoff statistic $W_j = w_j([X, \tilde{X}], Y)$ for $j \in [p]$, must satisfy the flip-sign property:

$$w_j \left( (X, \tilde{X})_{\text{swap}(B)}, Y \right) = \begin{cases} w_j((X, \tilde{X}), Y) \text{ if } j \notin B \\ -w_j((X, \tilde{X}), Y) \text{ if } j \in B \end{cases} \tag{2}$$

The functions $w_j(\cdot), j \in [p]$ have many candidates, for example $w_j = |\beta_j| - |\tilde{\beta}_j|$, where $\beta_j$ and $\tilde{\beta}_j$ are the corresponding regression coefficient of $X_j$ and $\tilde{X}_j$ with the regression function $Y \sim (X, \tilde{X})$.

When the two knockoff conditions (i.e. Eq. (1) and (2)) are met, one can select features by $S = \{W_j \geqslant \tau_q\}$, where

$$\tau_q = \min_{t > 0} \left\{ t : \frac{1 + |\{j : W_j \leqslant -t\}|}{\max(1, |\{j : W_j \geqslant t\}|)} \leqslant q \right\}. \tag{3}$$

To assess the feature selection quality, FDR (see Appendix A for precise definition) is commonly used as an average Type I error of selected features (Barber & Candès, 2015). The control of FDR is guaranteed by the following theorem from Candes et al. (2018):

**Theorem 1** *Given knockoff copy and knockoff statistic satisfying the aforementioned property, and $S = \{W_j \geqslant \tau_q\}$, we have* FDR $\leqslant q$.

### 2.2 RELATED WORKS

Model-X knockoff is first studied under Gaussian design. Namely, the original variable $X \sim \mathcal{N}(\mu, \Sigma)$ with $\mu$ and $\Sigma$ known. Since Gaussian design does not naturally generalize to complex data distributions, a number of methods are proposed to weaken the assumption. Among them, model-specific ones such as AEknockoff (Liu & Zheng, 2018) Hidden Markov Model (HMM) knockoff (Sesia et al., 2017), and MASS (Gimenez et al., 2019) all propose parametric alternatives to Gaussian design. These methods are able to better learn the data distribution, while keeping the sampling process relatively simple. Nevertheless, they pose assumptions to the design distribution, which can be problematic if actual data does not coincide. To gain further flexibility, various deep-learning-based models are developed to generate knockoffs from distributions beyond parametric setup. DDLK (Sudarshan et al., 2020) and sRMMD (Masud et al., 2021) utilize different metrics to measure the distances between the original and the knockoff covariates. They apply different regularization terms to impose the "swap property". He et al. (2021) proposes a KnockoffScreen procedure to generate multiple knockoffs to improve the stability by minimizing the variance during knockoff construction. KnockoffGAN (Jordon et al., 2018) and Deep Knockoff (Romano et al., 2020) take advantage of the deep learning structures to create likelihood-free generative models for the knockoff generation.

Despite flexibility to learn the data distribution, deep-learning-based models suffer from major drawbacks. Knockoff generations based on distribution-free sampling methods such as generative adversarial networks (GAN) (Goodfellow et al., 2020; Arjovsky et al., 2017) tend to overfit, namely to learn the data $X$ exactly. The reason is that the notion of swap property for continuous distributions is ill-defined on sample level. To satisfy swap property, one needs to independently sample $\tilde{X}_j$ from the conditional law $P_{X_j}(\cdot|X_{-j})$, where $X_{-j}$ denotes the vector $(X_1, \ldots, X_{j-1}, X_{j+1}, \ldots, X_p)$. On sample level, each realization of $X_{-j} = x^i_{-j}$ is almost surely different and only associates to one corresponding sample $X_j = x^i_j$, causing the conditional law to degenerate to sum of Diracs. As a result, any effort to minimize the distance between $(X, \tilde{X})$ and $(X, \tilde{X})_{\mathrm{swap}(B)}$ will push $\tilde{X}$ towards $X$ and introduce high collinearity that makes the feature selection powerless. To tackle the issue, DDLK (Sudarshan et al., 2020) suggests an entropic regularization. Yet it still lacks power and is computationally expensive.

### 2.3 BOOST POWER BY REDUCING RECONSTRUCTABILITY

The issue of lacking power in the knockoff selection is solved in the Gaussian case. Assuming the knowledge of both mean and covariance of $X \sim \mathcal{N}(\mu, \Sigma)$, the swap property is easily satisfied by setting $\tilde{X}_j \sim \mathcal{N}(\mu_j, \Sigma_{jj})$ and $\Sigma_{ij} = \mathrm{Var}(X_i, \tilde{X}_j)$, for $i \neq j$, $i, j \in [p]$. Barber & Candès (2015) originally propose to minimize $\mathrm{Var}(X_j, \tilde{X}_j)$ for all $j \in [p]$ using semi-definite programming (SDP), to prevent $\tilde{X}_j$ to be highly correlated with $X_j$. However, Spector & Janson (2022) observed that the SDP knockoff still lacks feature selection power, as merely decorrelating $X_j$ and $\tilde{X}_j$ is not enough, and $(X, \tilde{X})$ can still be (almost) linearly dependent in various cases. This is referred to as high reconstructability in their paper, which can be considered as a population counterpart of collinearity (see Appendix B for more details). To tackle the problem, Spector & Janson (2022) proposed to maximize the expected conditional variance $\mathbb{E}\mathrm{Var}(X_j \mid X_{-j}, \tilde{X})$, which admits close-form solution whenever $X$ is Gaussian.

## 3 METHOD

DeepDRK provides a novel way to generate knockoffs $\tilde{X}$ while reducing the reconstructability (see Section 2.3) between the generated knockoff $\tilde{X}$ and the input $X$ for data with complex distributions. Note that in this section we slightly abuse the notation such that $X$ and $\tilde{X}$ denote the corresponding data matrices. The generated knockoff can then be used to perform FDR-controlled feature selection following the Model-X knockoff framework (see Section 2.1). Overall, DeepDRK is a pipeline of two components. It first trains a transformer-based deep learning model, denoted as Knockoff Transformer (KT), to obtain swap property as well as reduce recontructability for the generated knockoff, with the help of adversarial attacks via multi-swappers. Second, a dependency regularized perturbation technique (DRP) is developed to further reduce the reconstructability for $\tilde{X}$ post training. In the following subsections, we discuss two components in detail in the following two subsections.

**DeepDRK Pipeline**

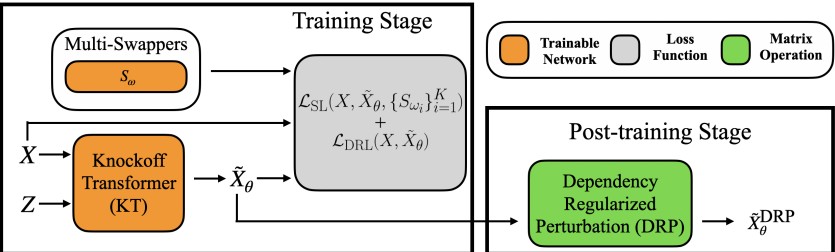

Figure 1: The diagram that illustrates the DeepDRK pipeline, which consists of two components: 1. the training stage that optimizes the knockoff Transformer and swappers by $\mathcal{L}_{\mathrm{SL}}$ and $\mathcal{L}_{\mathrm{DRL}}$; 2. the post-training stage that generates the knockoff $\tilde{X}^{\mathrm{DRP}_\theta}$ via dependency regularized perturbation.

### 3.1 TRAINING WITH KNOCKOFF TRANSFORMER AND SWAPPERS

The KT (i.e. $\tilde{X}_\theta$), shown in Fig. 1, is trained to generate knockoffs by minimizing a swap loss (SL) $\mathcal{L}_{\text{SL}}$ and a dependency regularization loss (DRL) $\mathcal{L}_{\text{DRL}}$ to achieve swap property and boost power. Swappers (i.e. $S_{\omega_i}$), on the other hand, are trained to generate adversarial swaps to enforce swap property. The total training objective is defined as

$$\min_\theta \max_\omega \left\{ \mathcal{L}_{\text{SL}}(X, \tilde{X}_\theta, \{S_{\omega_i}\}_{i=1}^K) + \mathcal{L}_{\text{DRL}}(X, \tilde{X}_\theta) \right\}. \tag{4}$$

Note that we use $\tilde{X}_\theta$ and $\tilde{X}$ interchangeably, whereas the former is to emphasize that the knockoff depends on the model weights $\theta$ for KT. We detail the design of KT and swappers in Appendix C. In the following subsections, we breakdown each loss function.

#### 3.1.1 SWAP LOSS

The swap loss is designed to enforce the swap property and is defined as follow:

$$\mathcal{L}_{\text{SL}}(X, \tilde{X}_\theta, \{S_{\omega_i}\}_{i=1}^K) = \frac{1}{K} \sum_{i=1}^K \text{SWD}([X, \tilde{X}_\theta], [X, \tilde{X}_\theta]_{S_{\omega_i}}) \tag{5}$$
$$+ \lambda_1 \cdot \text{REx}(X, \tilde{X}_\theta, \{S_{\omega_i}\}_{i=1}^K) + \lambda_2 \cdot \mathcal{L}_{\text{swapper}}(\{S_{\omega_i}\}_{i=1}^K)$$

where $\lambda_1$ and $\lambda_2$ are hyperparameters.

The first term in Eq. (5)—"SWD"—measures the distance between two corresponding distributions. In our case, they are the laws of the joint distributions $[X, \tilde{X}_\theta]$ and $[X, \tilde{X}_\theta]_{S_{\omega_i}}$. Here $\text{SWD}(\cdot, \cdot)$ is the sliced-Wasserstein distance (see Appendix D), $S_{\omega_i}$ is the $i$-th swapper with model parameters $\omega$, and $K$ is the total number of swappers. The swapper, introduced in (Sudarshan et al., 2020), is a neural network for generating adversarial swaps (i.e. attacks). Training against such attacks enforces the swap property (see Appendix C for a detailed description). We adopt the same idea and extend it to a "multi-swapper" setup that jointly optimizes multiple independent swappers to better achieve swap property. In comparison with the likelihood-based loss in (Sudarshan et al., 2020), which requires a normalizing flow (NF) (Papamakarios et al., 2021) to fit, SWD is adopted to compare distributions, as it not only demonstrates high performance on complex structures such as multi-modal data, but also fast to implement since it is likelihood-free (Kolouri et al., 2019; Deshpande et al., 2019; 2018; Nguyen et al., 2022).

The second term in Eq. (5)—"REx"—stands for risk extrapolation (Krueger et al., 2021) and is defined here as $\text{REx}(X, \tilde{X}_\theta, \{S_{\omega_i}\}_{i=1}^K) = \widehat{\text{Var}}_{S_{\omega_i}}(\text{SWD}([X, \tilde{X}_\theta], [X, \tilde{X}_\theta]_{S_{\omega_i}})$ for $i \in [K]$, which is a variant of the invariant risk minimization (IRM) (Arjovsky et al., 2019) loss. This term is commonly used to add robustness against adversarial attacks in generative modeling, which in our case, guarantees that the generated knockoffs $\tilde{X}_\theta$ are robust to multiple swap attacks simultaneously. Since swappers $S_{\omega_i}$ may have the same effect (identical adversarial attacks) if they share the same weights $\omega$, it is important to distinguish them by enforcing different pairs of weights via the third term in Eq. (5):

$$\mathcal{L}_{\text{swapper}}(\{S_{\omega_i}\}_{i=1}^K) = \frac{1}{|C|} \sum_{(i,j) \in C} \text{sim}(S_{\omega_i}, S_{\omega_j}), \tag{6}$$

where $C = \{(i,j) | i, j \in [K], i \neq j\}$, and $\text{sim}(\cdot, \cdot)$ is the cosine similarity function.

In all, the swap loss $\mathcal{L}_{\text{SL}}$ enforces the swap property via SWD and the novel multi-swapper design. Such design provides a better guarantee of the swap property through multiple adversarial swap attacks, which is shown in Appendix L.3 for details.

#### 3.1.2 DEPENDENCY REGULARIZATION LOSS

As discussed in Section 2.2, pursuing the swap property on sample level often leads to severe over-fitting of $\tilde{X}_\theta$ (i.e., pushing $\tilde{X}_\theta$ towards $X$), which results in high collinearity in feature selection. To remedy, the DRL is introduced to reduce the reconstructability between $X$ and $\tilde{X}$:

$$\mathcal{L}_{\text{DRL}}(X, \tilde{X}_\theta) = \lambda_3 \cdot \text{SWC}(X, \tilde{X}_\theta) + \lambda_4 \cdot \mathcal{L}_{\text{ED}}(X, \tilde{X}_\theta), \tag{7}$$

where $\lambda_3$ and $\lambda_4$ are hyperparameters. The SWC term in Eq. (7) refers to the sliced-Wasserstein correltaion (Li et al., 2023), which quantitatively measures the dependency between two random vectors in the same space. More specifically, let $Z_1$ and $Z_2$ be two $p$-dimensional random vectors. $\mathrm{SWC}(Z_1, Z_2) = 0$ indicates that $Z_1$ and $Z_2$ are independent, while $\mathrm{SWC}(Z_1, Z_2) = 1$ suggests a linear relationship between each other (see Appendix E for more details of SWC). In DeepDRK, we minimize SWC to reduce the reconstructability, a procedure similar to Spector & Janson (2022). The intuition is as follows. If the joint distribution of $X$ is known, then for each $j \in [p]$, the knockoff $\tilde{X}_j$ should be independently sampled from $p_j(\cdot|X_{-j})$. In such case the swap property is ensured, and collinearity is avoided due to independence. As we do not have access to the joint law, we want the variables to be less dependent. Since collinearity exists with in $X$, merely decorrelate $X_j$ and $\tilde{X}_j$ is not enough. Thus we minimize SWC so that $X$ and $\tilde{X}$ are less dependent. We refer readers to Appendix B for further discussion. In addition to SWC, the entry-wise decorrelation term $\mathcal{L}_{\mathrm{ED}}(X, \tilde{X}_\theta) = \sum_{j=1}^p \widehat{\mathrm{corr}}\left(X_j, \tilde{X}_{\theta,j}\right)$, proposed in Romano et al. (2020), could be optionally added to stabilize training. We consider this term for all the experiments considered in this paper, however, we recommend its removal for the general use of DeepDRK. The discussion on this is in Appendix F.

## 3.2 DEPENDENCY REGULARIZATION PERTURBATION

Empirically we observe a competition between $\mathcal{L}_{\mathrm{SL}}$ and $\mathcal{L}_{\mathrm{DRL}}$ in Eq. (4), which adds difficulty to the training procedure. Specifically, the $\mathcal{L}_{\mathrm{SL}}$ is dominating and the $\mathcal{L}_{\mathrm{DRL}}$ increases quickly after a short decreasing period. The phenomenon is also observed in all deep-learning based knockoff generation models when one tries to gain power (Romano et al., 2020; Sudarshan et al., 2020; Masud et al., 2021; Jordon et al., 2018). We inlcude the experimental evidence in Appendix G. We speculate the reasons as follows: minimizing the swap loss, which corresponds to FDR control, is the same as controlling Type I error. Similarly, minimizing the dependency loss is to control Type II error. With a fixed number of observations, it is well known that Type I error and Type II error can not decrease at the same time after reaching a certain threshold. In the framework of model-X knockoff, we aim to boost as much power as possible given the FDR is controlled at a certain level, a similar idea as the uniformly most powerful (UMP) test Casella & Berger (2021). For this reason, we propose DRP as a post-training technique to further boost power.

DRP is a sample-level perturbation that further eliminates dependency between $X$ and and the knockoff. More specifically, DRP perturbs the generated $\tilde{X}_\theta$ with the row-permuted version of $X$, denoted as $X_{\mathrm{rp}}$. After applying DRP, the final knockoff $\tilde{X}_\theta^{\mathrm{DRP}}$ becomes:

$$\tilde{X}_\theta^{\mathrm{DRP}} = \alpha \times \tilde{X}_\theta + (1 - \alpha) \times X_{\mathrm{rp}}, \tag{8}$$

where $\alpha$ is a preset perturbation weight. In this way $\tilde{X}^{\mathrm{DRP}}$ has a smaller SWC with $X$, since $X_{\mathrm{rp}}$ is independent of $X$. Apparently by the perturbation, swap loss increases. We show in Appendix J and L.2 the effect of the perturbation on the swap property, FDR, and power.

## 4 EXPERIMENT

We compare the performance of our proposed DeepDRK with a set of benchmarking models from three different perspectives. Namely, we focus on: 1. the fully synthetic setup that both input variables and the response variable have pre-defined distributions; 2. the semi-synthetic setup where input variables are rooted in real-world datasets and the response variable is created according to some known relationships with the input; 3. FS with a real-world dataset. Overall, the experiments are designed to cover various datasets with different $p/n$ ratios and different distributions of $X$, aiming to evaluate the model performance comprehensively. In the following, we first introduce the model and training setups for DeepDRK and the benchmarking models. Next we discuss the configuration of the feature selection. Finally, we describe the dataset setups, and the associated results are presented. Additionally, we consider the ablation study to illustrate the benefits obtained by having the following proposed terms: SWC, REx, $\mathcal{L}_{\mathrm{swapper}}$, and the multi-swapper setup. Empirically, these terms improve power and enforce the swap property for a lower FDR. We also demonstrate the improved stability by introducing entry-wise decorrelation $\mathcal{L}_{\mathrm{ED}}$. Due to space limitation, we defer details to Appendix L.3.

### 4.1 MODEL TRAINING & CONFIGURATION

To fit models, we first split datasets of $X$ into two parts, with an 8:2 training and validation ratios. The training sets are then used for model optimization, and validation sets are used for stopping the model training with an early stop rule on the validation loss with the patient period equals 6. Since knockoffs are not unique (Candes et al., 2018), there is no need of the testing sets. To evaluate the model performance of DeepDRK, we compare it with 4 state-of-the-art (SOTA) deep-learning-based knockoff generation models as we focus on non-parametric data in this paper. Namely, we consider Deep Knockoff (Romano et al., 2020) [2], DDLK (Sudarshan et al., 2020) [3], KnockoffGAN (Jordon et al., 2018) [4] and sRMMD (Masud et al., 2021) [5], with links of the code implementation listed in the footnote. We follow all the recommended hyperparameter settings for training and evaluation in their associated code repositories. The only difference is the total number of training epochs. To maintain consistency, we set this number to 200 for all models (including DeepDRK).

We follow the model configuration (see Appendix H) to optimize DeepDRK. The architecture of the swappers $S_\omega$ is adopted from Sudarshan et al. (2020). Optimizers for training both swappers and $\tilde{X}_\theta$ are AdamW (Loshchilov & Hutter, 2017). We interchangeably optimize for $\tilde{X}_\theta$ and swappers $S_\omega$ during training (see Eq. (4)). For every three times of updates for weights $\theta$, the weights $\omega$ are updated once. The training scheme is similar to the training of GAN (Goodfellow et al., 2020), except the absence of discriminators. We also apply early stopping criteria to prevent overfitting. A pseudo code for the optimization can be found in Appendix I. To conduct experiments, we set $\alpha = 0.5$ universally for the dependency regularization coefficient as it leads to consistent performance across all experiments. A discussion on the effect of $\alpha$ is detailed in Appendix J. Additionally, we consider the effect of model size to the FS performance in Appendix K.

Once models are trained, we generate the knockoff $\tilde{X}$ given the data of $X$. The generated $\tilde{X}$ are then combined with $X$ for feature selection. We run each experiment on a single NVIDIA V100 16GB GPU. DeepDRK is implemented by `PyTorch` (Paszke et al., 2019) [6].

### 4.2 FEATURE SELECTION CONFIGURATION

Once the knockoff $\tilde{X}$ is obtained, feature selection is performed following three steps. First, concatenate $[X, \tilde{X}]$ to form a $n \times (2p)$ design matrix $(X, \tilde{X})$. Second, fit a regression model $Y \sim (X, \tilde{X})$ to obtain estimation coefficients $\{\hat{\beta}_j\}_{j \in [2p]}$. We use Ridge regression as it generally results in higher power. Third, compute knockoff statistics $W_j = |\hat{\beta}_j| - |\hat{\beta}_{j+p}|$ (for $j = 1, 2, \ldots, p$), and select features according to Eq. (3). We consider $q = 0.1$ as the FDR threshold due to its wide application in the analysis by other knockoff-based feature selection papers (Romano et al., 2020; Masud et al., 2021). Each experiment is repeat 600 times, and the following results with synthetic and semi-synthetic datasets are provided with the average power and FDR of the 600 repeats.

### 4.3 THE SYNTHETIC EXPERIMENTS

To properly evaluate the performance, we follow a well-designed experimental setup by Sudarshan et al. (2020); Masud et al. (2021) to generate different sets of $n$-samples of $(X, Y)$. Here $X \in \mathbb{R}^p$ is the collection dependent variables that follows pre-defined distributions. $Y \in \mathbb{R}$ is the response variable that is modeled as $Y \sim \mathcal{N}(X^T \beta, 1)$. The underlying true $\beta$ is the $p$-dimensional coefficient which is set to follow $\frac{p}{15 \cdot \sqrt{n}} \cdot$ Rademacher(0.5) or $\frac{p}{12.5 \cdot \sqrt{n}} \cdot$ Rademacher(0.5) distribution. Compared to Sudarshan et al. (2020) and Masud et al. (2021), which consider $\frac{p}{\sqrt{n}}$ as the scalar for the Rademacher, we downscale the scale of $\beta$ by the factors of 15 or 12.5. This is because we find that in the original setup, the $\beta$ scale is too large such that the feature selection enjoys high powers and low FDRs for all models. To compare the performance of the knockoff generation methods on various data, we consider the following distributions for $X$:

---

[2] https://github.com/msesia/deepknockoffs

[3] https://github.com/rajesh-lab/ddlk

[4] https://github.com/vanderschaarlab/mlforhealthlabpub/tree/main/alg/knockoffgan

[5] https://github.com/ShoaibBinMasud/soft-rank-energy-and-applications

[6] The implementation of DeepDRK can be found in https://NODISCLOSUREFORPEERREVIEW.

`Mixture of Gaussians`: Following Sudarshan et al. (2020), we consider a Gaussian mixture model $X \sim \sum_{k=1}^{3} \pi_k \mathcal{N}(\mu_k, \Sigma_k)$, where $\pi$ is the proportion of the $k$-th Gaussian with $(\pi_1, \pi_2, \pi_3) = (0.4, 0.2, 0.4)$. $\mu_k \in \mathbb{R}^p$ denotes the mean of the $k$-th Gaussian with $\mu_k = \mathbf{1}_p \cdot 20 \cdot (k-1)$, where $\mathbf{1}_p$ is the $p$-dimensional vector that has universal value 1 for all entries. $\Sigma_k \in \mathbb{R}^{p \times p}$ is the covariance matrices whose $(i, j)$-th entry taking the value $\rho_k^{|i-j|}$, where $(\rho_1, \rho_2, \rho_3) = (0.6, 0.4, 0.2)$.

`Copulas` (Schmidt, 2007): we further use copula to model complex correlations within $X$. To our knowledge, this is a first attempt to consider complicated distributions other than the mixture Gaussian in the knockoff framework. Specifically, we consider two copula families: Clayton, Joe with the consistent copula parameter of 2 in both cases. For each family, two candidates of the marginal distributions are considered: uniform (i.e. identity conversion function) and the exponential distribution with rate equals 1. We implement copulas according to `PyCop` [7].

For both cases above, we consider the following $(n, p)$ setups in the form of tuples: $(200, 100)$, $(500, 100)$, and $(2000, 100)$. This is in contrast to existing works, which considers only the $(2000, 100)$ case for small $n$ value (Sudarshan et al., 2020; Masud et al., 2021; Romano et al., 2020). Our goal is to demonstrate a consistent performance by the proposed DeepDRK on different $p/n$ ratios, especially when it is small.

`Results`: Figure 2 and 3 compare FDRs and Powers for all models across the previously introduced datasets with two different setups for $\beta$, respectively. For illustration purposes, we keep sample size 200 and 2000 cases and defer the sample size 500 case to Appendix L.1. It is observed in Figure 2 that DeepDRK is capable of controlling FDR consistently compared to other benchmarking models across different data distributions and different $p/n$ ratios. Other models, though being able to reach higher power, comes at a cost of sacrificing FDR, which contradicts to the UMP philosophy (see Section 3.2). The results in Figure 3 share a similar pattern to that of Figure 2, except for the number of FDR-inflated cases (i.e. above the 0.1 threshold) increases due to the lowered scale of $\beta$. Overall, the results demonstrate the ability of DeepDRK in consistently performing FS with controlled FDR compared to other models across a range of different datasets and $p/n$ ratios. Additionally, we provide the measurement of the swap property and the average runtime for each model in Appendix L.2 and M, respectively.

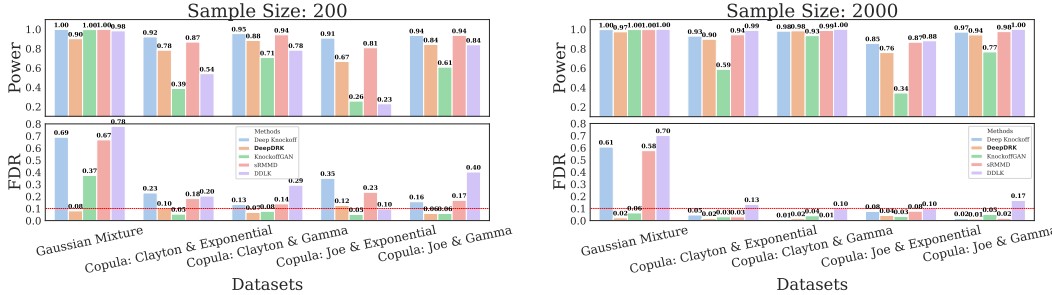

Figure 2: FDR and power comparison across different models with the synthetic datasets with $\beta \sim \frac{p}{12.5 \cdot \sqrt{N}} \cdot \text{Rademacher}(0.5)$. The red horizontal bar indicates the threshold $q$ during FS.

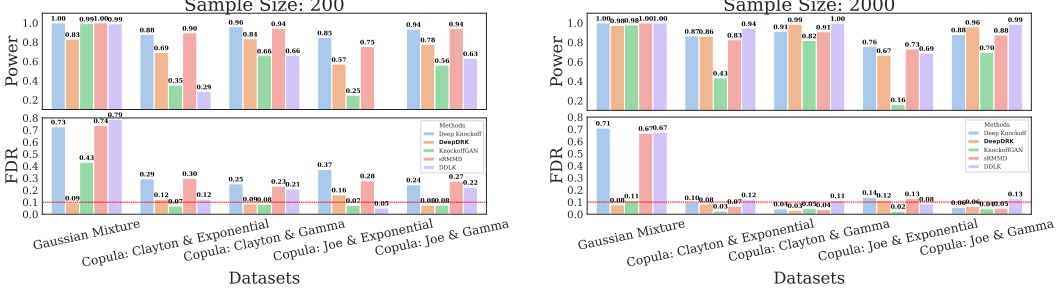

Figure 3: FDR and power comparison across different models with the synthetic datasets with $\beta \sim \frac{p}{15 \cdot \sqrt{N}} \cdot \text{Rademacher}(0.5)$. The red horizontal bar indicates the threshold $q$ during FS.

[7] https://github.com/maximenc/pycop/

### 4.4 THE SEMI-SYNTHETIC EXPERIMENTS

Following Hansen et al. (2022) and Sudarshan et al. (2020), we consider a semi-synthetic study with design $X$ drawn from two real-world datasets, and use $X$ to simulate response $Y$. The details of which are as follows:

The first dataset contains single-cell RNA sequencing (scRNA-seq) data from 10x Genomics [8]. Each entry in $X \in \mathbb{R}^{n \times p}$ represents the observed gene expression of gene $p$ in cell $n$. We refer readers to Hansen et al. (2022) and Agarwal et al. (2020) for further background. Following the same preprocessing in Hansen et al. (2022), we obtain the final dataset of $X$ is of size 10000 and dimension 100 [9]. The preprocessing of $X$ and the synthesis of $Y$ are deferred in Appendix N.

The second publicly available[10] dataset originates from a real case study entitled "Longitudinal Metabolomics of the Human Microbiome in Inflammatory Bowel Disease (IBD)" (Lloyd-Price et al., 2019). The study seeks to identify important metabolites of two representative diseases of the inflammatory bowel disease (IBD): ulcerative colitis (UC) and Crohn's disease (CD). Specifically, we use the C18 Reverse-Phase Negative Mode dataset that has 546 samples, each of which has 91 metabolites (i.e. $X$). To mitigate the effects of missing values, we preprocess the dataset following a common procedure to remove metabolites that have over 20% missing values, resulting in 80 metabolites, which are then normalized to have zero mean and unit variance after a log transform and an imputation via the k nearest neighbor algorithm following the same procedure in Masud et al. (2021). Finally, we synthesize the response $Y$ with the real dataset of $X$ via $Y \sim \mathcal{N}(X^T \beta, 1)$, where $\beta$ follows entry-wise Unif(0, 1), $\mathcal{N}(0, 1)$, and Rademacher(0.5) distributions.

`Results`: Figure 4 and 5 compare the feature selection performance on the RNA data and the IBD data respectively. In Figure 4, we observe that all but DDLK are bounded by the nominal 0.1 FDR threshold in the "Tanh" case. However, KnockoffGAN and sRMMD have almost zero power. The power for Deep Knockoff is also very low compared to that of DeepDRK. Although DDLK provides high power, the associated FDR is undesirable. In the "Linear" case, almost all models have well controlled FDR, among which DeepDRK provides the highest power. Similar observations can be found in Figure 5. For the IBD data under the aforementioned three different data synthesis rules, it is clear all models but DDLK achieve well-controlled FDR. Apart from DDLK, DeepDRK universally achieves the highest powers. The results further demonstrate the potential of DeepDRK on real data applications.

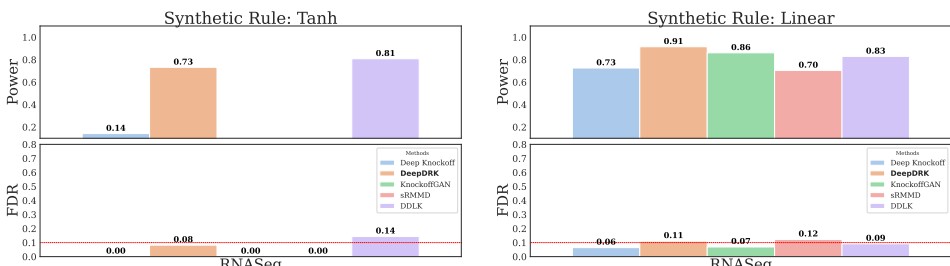

Figure 4: FDR and power comparison across different models with the semi-synthetic RNA dataset. The red horizontal bar indicates the threshold $q$ during FS.

### 4.5 A CASE STUDY

Besides (semi-)synthetic setups, we follow (Sudarshan et al., 2020; Romano et al., 2020) and carry out a case study with real data for both design $X$ and response $Y$, in order to qualitatively evaluate the selection performance of DeepDRK. In this subsection, we consider the full IBD dataset (i.e. with both $X$ and $Y$ from the dataset (Lloyd-Price et al., 2019)). The response variable $Y$ is categorical: $Y$ equals 1 if a given sample is associated with UC/CD and 0 otherwise. The covariates $X$ is identical to the second semi-synthetic setup considered in Section 4.4. To properly evaluate results

---

[8] `https://kb.10xgenomics.com/hc/en-us`

[9] The data processing code is adopted from this repo: `https://github.com/dereklhansen/flowselect/tree/master/data`

[10] `https://www.metabolomicsworkbench.org/` under the project DOI: 10.21228/M82T15.

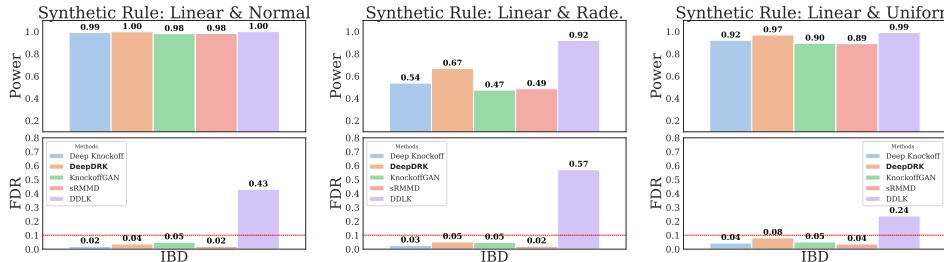

Figure 5: FDR and power comparison across different models with the semi-synthetic IBD dataset. The red horizontal bar indicates the threshold $q$ during FS.

| Model | DeepDRK | Deep Knockoff | sRMMD | KnockoffGAN | DDLK |
|---|---|---|---|---|---|
| Referenced / Identified | 37/52 | 15/20 | 5/5 | 12/14 | 17/25 |

Table 1: The number of identified metabolites v.s. the number of total discovery in the IBD dataset.

with no ground truth available, we investigate literature with different sources, searching for the evidence of the IBD-associated metabolites. Namely, we curate three sources: 1. metabolites that are explicitly documented to have associations with IBD, UC, or CD in the PubChem database [11]; 2. metabolites that are reported in the existing peer-reviewed publications; 3. metabolites that are reported in pre-prints. To our knowledge, we are the first to carry out a comprehensive metabolite investigation for the IBD dataset. All referenced metabolites are included in Table 13 in Appendix O. In all, we identify 47 metabolites that are reported to have association with IBD.

To evaluate model performance, we first consider the same setup (see Table 2 in Appendix H) to train the model and generate knockoffs using the DeepDRK pipeline. During the FS step, however, we use 0.2 as the FDR threshold instead of 0.1, and apply a different algorithm—DeepPINK (Lu et al., 2018) that is included in the `knockpy` [12] library—to generate knockoff statistics $W_j$. The values are subsequently used to identify metabolites. We choose Deep Pink—a neural-network-based model–over the previously considered ridge regression due to the nonlinear relationships between metabolites (i.e. $X$) and labels (i.e. $Y$) in this case study. Likewise, to generate knockoff for the benchmarking models, we follow their default setup. During FS, same changes are applied as in DeepDRK.

We compare the FS results with the 47 metabolites and report the number of selections in Table 1. A detailed list of selected features for each model can be found in Table 14 in Appendix O. From Table 1, we clearly observe that compared to the benchmarking models, DeepDRK identifies the most number of referenced metabolites. KnockoffGAN and sRMMD behave preferably in limiting the number of discoveries given the fewer number of identified undocumented metabolites. The performance of Deep Knockoff and DDLK is intermediate, meaning a relatively higher number of discovered metabolites that are referenced, compared to KnockoffGAN and sRMMD, and a relatively lower number of discovered undocumented metabolites. Nevertheless, since there is no ground truth available, the comparison is only qualitative.

## 5 CONCLUSION

In this paper, we introduced DeepDRK, a deep learning-based knockoff generation pipeline that consists of two steps. It first trains a Knockoff Transformer with multi-swappers to obtain swap property as well as reduce reconstructability. Second, the dependency regularized perturbation is applied to further boost power after training. DeepDRK is shown to achieve both low FDR and high power across various data distributions, with different $p/n$ ratios. Our results also suggest an outperformance of DeepDRK over all considered deep-learning-based benchmarking models. Experiments with real and semi-synthetic data further demonstrate the potential of DeepDRK in feature selection tasks with complex data structures.

---

[11]https://pubchem.ncbi.nlm.nih.gov/
[12]https://amspector100.github.io/knockpy/

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

APPENDIX

## A  FDR DEFINITION

The definition of FDR is a follows. Let $S \subset [p]$ be any selected indices and $\beta^*$ be the underlying true regression coefficients. The FDR for selection $S$ is

$$\text{FDR} = \mathbb{E}\left[\frac{\#\left\{j : \beta_j^* = 0 \text{ and } j \in S\right\}}{\#\{j : j \in S\} \vee 1}\right] \tag{9}$$

## B  RECONSTRUCTABILITY AND SELECTION POWER

The notion of reconstructability is introduced by Spector & Janson (2022) as a population level counterpart of what is usually referred to as collinearity in linear regression. Under Gaussian design where $X \sim \mathcal{N}(0, \Sigma)$, reconstructability is high if $\Sigma$ is not of fill rank, so that $\exists j$ s.t. $X_j$ is a.s. a linear combination of $X_{-j}$. More generally, if there exists more than one representation of the response $Y$ using the explanatory variable $X$, we qualitatively say that the reconstructability is high. As high collinearity often causes trouble, reconstructability also hurts power in feature selection. To better illustrate, we state a linear version of Theorem 2.3 in Spector & Janson (2022), which is originally a more general single-index model (Hristache et al., 2001).

**Theorem 2** *Let $Y = X_J \beta_J + X_{-J} \beta_{-J} + \varepsilon$, where $J \subset [p]$, and $\varepsilon$ is a centered Gaussian noise. Equivalently it means $Y \perp\!\!\!\perp X_J \mid X_J \beta_J, X_{-J}$. Suppose there exists a $\beta_J^*$ such that $X_J \beta_J = X_J \beta_J^*$ a.s., then denoting $Y^* = X_J \beta_J^* + X_{-J} \beta_{-J} + \varepsilon$, we have*

$$\left([X, \tilde{X}], Y\right) \stackrel{d}{=} \left([X, \tilde{X}]_{\text{swap}(J)}, Y^*\right) \quad \text{and} \quad \left([X, \tilde{X}], Y^*\right) \stackrel{d}{=} \left([X, \tilde{X}]_{\text{swap}(J)}, Y\right). \tag{10}$$

*Furthermore in the knockoff framework (Barber & Candès, 2015), let $W = w([X, \tilde{X}], y)$ and $W^* = w([X, \tilde{X}], y^*)$, then for all $j \in J$,*

$$\mathbb{P}\left(W_j > 0\right) + \mathbb{P}\left(W_j^* > 0\right) \leqslant 1. \tag{11}$$

Equation 11 implies a no free lunch situation for selection power when there is exact reconstructability.

To fix the reconstructability issue, Spector & Janson (2022) proposed two methods in Gaussian design. The first is the minimal variance-based reconstructability (MVR) knockoff, in which knockoff $\tilde{X}$ is sampled to minimize the loss

$$L_{\text{MVR}} = \sum_{j=1}^{p} \frac{1}{\mathbb{E}\left[\text{Var}\left(X_j \mid X_{-j}, \tilde{X}\right)\right]}. \tag{12}$$

Note that is equivalent to maximize $\mathbb{E}\left[\text{Var}\left(X_j \mid X_{-j}, \tilde{X}\right)\right]$ for all $j \in [p]$. Another way to achieve it is the maximum entropy (ME) knockoff, where $\tilde{X}$ is sampled from maximizing

$$L_{\text{ME}} = \int \int p(x, \tilde{x}) \log(p(x, \tilde{x})) d\tilde{x} dx. \tag{13}$$

Fortunately under Gaussian design, the above two optimizations have closed-form solutions. Since $X$ is Gaussian, $(X, \tilde{X})$ must be joint Gaussian to satisfy the swap property. To optimize, one first calculates the covariance matrix using the SDP method in Barber & Candès (2015), which gives a diagonal matrix $S$. Then both MVR and ME boils down to an optimization on $S$:

$$L_{\text{MVR}}(S) \propto \text{Tr}\left(G_S^{-1}\right) = \sum_{j=1}^{2p} \frac{1}{\lambda_j(G_S)} \quad \text{and} \quad L_{\text{ME}}(S) = \log \det\left(G_S^{-1}\right) = \sum_{j=1}^{2p} \log\left(\frac{1}{\lambda_j(G_S)}\right). \tag{14}$$

Though it is shown that both methods obtain high power for feature selection, however, neither MVR nor ME could be directly extended to arbitrary distributions, due to the fact that the conditional variance and the likelihood is intractable.

In DeepDRK, we consider regularizing with a sliced-Wasserstein-based dependency correlation, which can be deemed a stronger dependency regularization than entropy. A post-training perturbation is also applied to further reduce collinearity. However, theoretical understanding of how they affect swap property and power remains to be studied.

## C   THE DESIGN OF KNOCKOFF TRANSFORMER AND SWAPPERS

DeepDRK's knockoff Transformer (KT) model is designed based on the popular Vision Transformer (ViT) (Dosovitskiy et al., 2020). The difference being the dimension of the input is 1D, not 2D, for $X$. We do not consider patches as the input. Instead, we consider all entries of $X$ to consider correlations of each pair of entries in the knockoff $\tilde{X}$. This is structurally similar to the original Transformer (Vaswani et al., 2017). Nonetheless, we keep all the remaining components from ViT, such as patch embedding, PreNorm and 1D-positional encoding (Dosovitskiy et al., 2020). Since knockoff generation essentially requires a distribution. To enable this, we consider feeding $X$ and a uniformly distributed random variable $Z$ of the same dimension as $X$, to encode randomness.

The swapper module is first introduced in DDLK (Sudarshan et al., 2020), to produce the index subset $B$ for the adversarial swap attack. Optimizing knockoff against these adversarial swaps enforces the swap property. Specifically, the swapper consists of a matrix of shape $2 \times p$ (i.e. trainable model weights), where $p$ is the dimension of $X$. This matrix controls the Gumbel-softmax distribution (Jang et al., 2017) for all $p$ entries. Each entry is characterized by a binary Gumbel-softmax random variable (e.g. can only take values of 0 or 1). To generate the subset $B$, we consider drawing samples of $b_j$ from the corresponding $j$-th Gumbel-softmax random variable, and the subset $B$ is defined as $\{j \in [p]; b_j = 1\}$. During optimization, we maximize Eq. (4) w.r.t. to the weights $\omega_i$ of the swapper $S_{\omega_i}$ such that the sampled indices, with which the swap is applied, lead to a higher SWD in the objective (Eq. (4)). Minimizing this objective w.r.t. $\tilde{X}_\theta$ requires the knockoff to fight against the adversarial swaps. Therefore, the swap property is enforced. Compared to DDLK, the proposed DeepDRK utilizes multiple independent swappers.

## D   FROM WASSERSTEIN TO SLICED-WASSERSTEIN DISTANCE

Wasserstein distance has become popular in both mathematics and machine learning due to its ability to compare different kinds of distributions (Villani et al., 2009) and almost everywhere differentiability (Arjovsky et al., 2017). Here we provide its definition. Let $X, Y$ be two $\mathbb{R}^d$ random vectors following distributions $\mathbf{P}_X, \mathbf{P}_Y$ with finite $p$-th moment. The *Wasserstein-p* distance between $\mathbf{P}_X$ and $\mathbf{P}_Y$ is:

$$W_p(\mathbf{P}_X, \mathbf{P}_Y) = \inf_{\gamma \in \Gamma(\mathbf{P}_X, \mathbf{P}_Y)} \left( \mathbb{E}_{(x,y) \sim \gamma} \|x - y\|^p \right)^{\frac{1}{p}} \tag{15}$$

where $\Gamma(\mathbf{P}_X, \mathbf{P}_Y)$ denotes the set of all joint distributions such that their marginals are $\mathbf{P}_X$ and $\mathbf{P}_Y$. When $d = 1$ in particular, the Wasserstein distance between two one-dimensional distributions can be written as:

$$W_p(\mathbf{P}_X, \mathbf{P}_Y) = \left( \int_0^1 |F_X^{-1}(v) - F_Y^{-1}(v)|^p dv \right)^{\frac{1}{p}} = \|F_X^{-1} - F_Y^{-1}\|_{L^p([0,1])}, \tag{16}$$

where $F_X$ and $F_Y$ are the cumulative distribution functions (CDF) of $\mathbf{P}_X$ and $\mathbf{P}_Y$ respectively. Moreover, if $p = 1$ as well, the Wasserstein distance can be further simplified as

$$W_1(\mathbf{P}_X, \mathbf{P}_Y) = \left( \int |F_X(v) - F_Y(v)| dv \right) = \|F_X - F_Y\|_{L^1(\mathbb{R})}. \tag{17}$$

From the above it is easy to notice that the $1d$ Wasserstein distance is easy to compute, which leads to the development of sliced-Wasserstein distance (SWD) (Bonneel et al., 2015). To exploit the computational advantage on $1d$, one first projects both distributions uniformly on a $1d$ direction and

computes the Wasserstein-$p$ distance between the two projected distributions. SWD is then calculated by taking the expectation of the random direction. More specifically, let $\mu \in \mathbb{S}^{d-1}$ denote a projection directions, the following push-forward distribution (Villani et al., 2009) $\mu_\sharp \mathbf{P}_X$ denotes the law of $\mu^T X$. Let $\mu$ be $d$ dimensional spherical uniform, the $p$-sliced-Wasserstein distance between $\mathbf{P}_X$ and $\mathbf{P}_Y$ reads:

$$SW_p(\mathbf{P}_X, \mathbf{P}_Y) = \int_{\mu \in \mathbb{S}^{d-1}} W_p(\mu_\sharp \mathbf{P}_X, \mu_\sharp \mathbf{P}_Y) \, d\mu. \tag{18}$$

Combining (18) and (16) gives:

$$SW_p(\mathbf{P}_X, \mathbf{P}_Y) =$$

$$\int_{\mu \in \mathbb{S}^{d-1}} \left( \int_0^1 |(F_X^\mu)^{-1}(v) - (F_Y^\mu)^{-1}(v)|^p dv \right)^{\frac{1}{p}} d\mu \tag{19}$$

$$= \int_{\mu \in \mathbb{S}^{d-1}} \int |F_X^\mu(v) - F_Y^\mu(v)| dv d\mu \quad when \, p = 1. \tag{20}$$

In spite of faster computations, the convergence of SWD is shown to be equivalent to the convergence of Wasserstein distance under mild condtions (Bonnotte, 2013). In practice, the expectation on $\mu$ is approximated by a finite summation over a number of projection directions uniformly chosen from $\mathbb{S}^{d-1}$.

## E   SLICED-WASSERSTEIN CORRELATION

The idea of metricizing independence is recently advanced using the Wasserstein distance (Wiesel, 2022; Nies et al., 2021). Given a joint distribution $(X, Y) \sim \Gamma_{XY}$ and its marginal distributions $X \sim \mathbf{P}_X, Y \sim \mathbf{P}_Y$, the Wasserstein Dependency (WD) between $X$ and $Y$ is defined by $WD(X, Y) = W_p(\Gamma_{XY}, \mathbf{P}_X \otimes \mathbf{P}_Y)$. A trivial observation is that $WD(X, Y) = 0$ implies that $X$ and $Y$ are independent. Due to the high computational cost of Wasserstein distance, sliced-Wasserstein dependency (SWDep) (Li et al., 2023) is developed using sliced-Wasserstein distance (see Appendix D for SWD details). The SW dependency between $X$ and $Y$ is defined as $SW_p(\Gamma_{XY}, \mathbf{P}_X \otimes \mathbf{P}_Y)$, and a 0 SW dependency still indicates independence. Since the dependency metric is not bounded from above, sliced-Wasserstein correlation (SWC) is introduced to normalize SW dependency. More specifically, the SWC between $X$ and $Y$ is defined as

$$\text{SWC}_p(X, Y) := \frac{\text{SWDep}_p(X, Y)}{\sqrt{\text{SWDep}_p(X, X) \, \text{SWDep}_p(Y, Y)}} \tag{21}$$

$$= \frac{\text{SWD}_p(\Gamma_X Y, \mathbf{P}_X \otimes \mathbf{P}_Y)}{\sqrt{\text{SWD}_p(\Gamma_{XX}, \mathbf{P}_X \otimes \mathbf{P}_X) \, \text{SWD}_p(\Gamma_{YY}, \mathbf{P}_Y \otimes \mathbf{P}_Y)}}, \tag{22}$$

where $\Gamma_X X$ and $\Gamma_Y Y$ are the joint distributions of $(X, X)$ and $(Y, Y)$ respectively. It is shown that $0 \leqslant \text{SWC}_p(X, Y) \leqslant 1$ and $\text{SWC}_p(X, Y) = 1$ when $X$ has a linear relationship with $Y$ (Li et al., 2023).

In terms of computing SWC, we follow Li et al. (2023) and consider both laws of $X$ and $Y$ to be sums of $2n$ Diracs, i.e. both variables are empirical distributions with data $\mathcal{I}_{\text{full}} = \{(\mathbf{x}_i, \mathbf{y}_i)\}_{i=1}^{2n}$. Define $\mathcal{I} = \{(\mathbf{x}_i, \mathbf{y}_i)\}_{i=1}^n$ and $\tilde{\mathcal{I}} = \{(\tilde{\mathbf{x}}_i, \tilde{\mathbf{y}}_i)\}_{i=1}^n$, where $(\tilde{\mathbf{x}}_i, \tilde{\mathbf{y}}_i) = (\mathbf{x}_{n+i}, \mathbf{y}_{n+i})$. Because data is i.i.d., $\mathcal{I}$ and $\tilde{\mathcal{I}}$ are independent. We further introduce the following notation:

$$\mathcal{I}_{\mathbf{xy}} = \{(\mathbf{x}_i, \mathbf{y}_i)\}_{i=1}^n, \tilde{\mathcal{I}}_{\mathbf{xy}} = \{(\tilde{\mathbf{x}}_i, \mathbf{y}_i)\}_{i=1}^n$$
$$\mathcal{I}_{\mathbf{xx}} = \{(\mathbf{x}_i, \mathbf{x}_i)\}_{i=1}^n, \tilde{\mathcal{I}}_{\mathbf{xx}} = \{(\tilde{\mathbf{x}}_i, \mathbf{x}_i)\}_{i=1}^n,$$
$$\mathcal{I}_{\mathbf{yy}} = \{(\mathbf{y}_i, \mathbf{y}_i)\}_{i=1}^n, \tilde{\mathcal{I}}_{\mathbf{yy}} = \{(\tilde{\mathbf{y}}_i, \mathbf{y}_i)\}_{i=1}^n$$

Then the empirical SWC can be computed by:

$$\widehat{\text{SWC}}_p(X, Y) := \frac{\text{SWD}_p\left(I_{\mathcal{I}_{\mathbf{xy}}}, I_{\tilde{\mathcal{I}}_{\mathbf{xy}}}\right)}{\sqrt{\text{SWD}_p\left(I_{\mathcal{I}_{\mathbf{xx}}}, I_{\tilde{\mathcal{I}}_{\mathbf{xx}}}\right) \text{SWD}_p\left(I_{\mathcal{I}_{\mathbf{yy}}}, I_{\tilde{\mathcal{I}}_{\mathbf{yy}}}\right)}}. \tag{23}$$

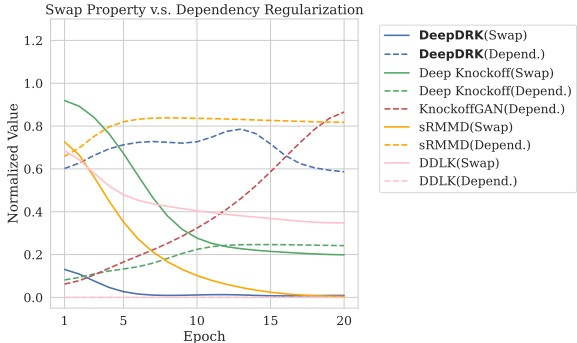

Figure 6: The competing relationship between the swap property (e.g. $\mathcal{L}_{\mathrm{SL}}$ in solid curves) and dependency regularization ($\mathcal{L}_{\mathrm{DRL}}$ in dashed curves).

## F    ENTRY-WISE DECORRELATION

The entry-wise decorrelation term between $X_j$ and $\tilde{X}_{\theta,j}$ in Eq. (7) is

$$\mathcal{L}_{\mathrm{ED}}(X, \tilde{X}_\theta) = \lambda_4 \cdot \sum_{j=1}^{p} \widehat{\mathrm{corr}}\left(X_j, \tilde{X}_{\theta,j}\right). \tag{24}$$

This term is adopted from Romano et al. (2020) for the purpose of stabilizing training in DeepDRK. We discover that the training of the knockoff transformer would occasionally experience instability issue, leaving the training loss to go infinity on some datasets. And adding this loss term prevents this situation. We conduct experiments to verify that the removal of this term does not significantly affect the performance of DeepDRK in feature selection. Results are deferred in the appendix L.3 for structural consistency. In DeepDRK, we include this term and the hyperparameter configuration for $\lambda_4$ can be found in Table 2. However, we suggest removing this term initially for reducing the search space of hyperparameters.

## G    COMPETING LOSSES

In Figure 6, we include scaled $\mathcal{L}_{\mathrm{SL}}$ curves and $\mathcal{L}_{\mathrm{DRL}}$ curves for each considered model. For comparison purposes, the curves that describe the changes of the corresponding losses are shown after normalizing the values to be within 0 and 1. We consider the first 20 epochs as the DRL would flatten out in latter epochs without dropping. The competition can be clearly observed as when $\mathcal{L}_{\mathrm{SL}}$ drops, $\mathcal{L}_{\mathrm{DRL}}$ increases, indicating difficulty to maintain low reconstructability.

## H    MODEL TRAINING CONFIGURATION

In Table 2, we include configuration details on the KT and swappers.

## I    TRAINING ALGORITHM

In Algorithm 1, we provide pseudo code for training the Knockoff Transformer (i.e. the first stage shown in Figure 1).

## J    EFFECT OF $\alpha$ IN $\tilde{X}_\theta^{\mathrm{DRP}}$

As discussed in section 3.2, empirically it is difficult to obtain swap property while maintaining low reconstructability at sample level. To leverage this issue, dependency regularization perturbation (DRP) is introduced. In this section, we evaluate the effect of $\alpha$ in $\tilde{X}_\theta^{\mathrm{DRP}}$ (e.g. Eq. (8)) to the feature selection performance. Results are summarized in Figure 7, concerning 5 synthetic datasets.

| Parameter | Value | Parameter | Value |
|---|---|---|---|
| # of $S_\omega$ | 2 | $S_\omega$ Temperature | 0.2 |
| $S_\omega$ Learning Rate | $1 \times 10^{-3}$ | $\tilde{X}_\theta$ Learning Rate | $1 \times 10^{-5}$ |
| Dropout Rate | 0.1 | # of Epochs | 200 |
| Batch Size | 64 | $\lambda_1$ | 30.0 |
| $\lambda_2$ | 1.0 | $\lambda_3$ | 20.0 |
| $\lambda_4$ | 10.0 | # of Layers | 8 |
| # of heads in $\tilde{X}_\theta$ | 8 | Hidden Dim | 512 |
| $\alpha$ | 0.5 | Early Stop Tolerance | 6 |

Table 2: Model setup and training Configuration. We remark that "$\lambda_4$" is associated with the optional term $\mathcal{L}_{\text{ED}}$ of Eq. (24).

When $\alpha$ is decreased, we observe an increment in power. However, the FDR follows a bowel-shaped pattern. This is consistent with the statement in Spector & Janson (2022), as introducing the permuted $X_{\text{rp}}$ will decrease reconstructability for higher power. However, a dominating $X_{\text{rp}}$ breaks the swap property for higher FDRs. Based on our hyperparameter search and the results in Figure 7, we suggest choosing $\alpha$ within the range between 0.4 and 0.5.

---

**Algorithm 1** Train the Knockoff Transformer (KT)

---

**Require:** Knockoff transformer $\tilde{X}_\theta$, denoted as $g_\theta(\cdot)$ for convenient; swappers $S_\omega$; number of swappers $K$; learning rate $\alpha_s$ for the swappers; learning rate $\alpha_\theta$ for the knockoff transformer; early stop tolerance $\eta$; number of epochs $T$; batch size $B_s$; dataset $\mathcal{D}$; Swapper update frequency $\gamma = 3$
**Ensure:** The output $\theta$ for $\tilde{X}_\theta$
1: **procedure** TRAINKT
2:      Split dataset $\mathcal{D}$ in to the training set $\mathcal{D}_{\text{train}}$ and the validation set $\mathcal{D}_{\text{val}}$        ▷ Initialization
3:      Initialize the knockoff transformer $g_\theta(\cdot)$ with random weights
4:      Initialize swappers $S_{\omega_i}$, $i = 1, \ldots, K$ with random weights
5:      Initialize the AdamW optimizer $\text{opt}_\theta$ with the learning rate $\alpha_\theta$ for $g_\theta(\cdot)$
6:      Initialize the AdamW optimizer $\text{opt}_{\omega_i}$ with the learning rate $\alpha_s$ for $S_{\omega_i}$, $i = 1, \ldots, K$
7:      **for** $t = 1$ **to** $T$ **do**        ▷ Training start
8:          **for** $l = 1$ **to** $\frac{|\mathcal{D}_{\text{train}}|}{B_s}$ **do**
9:              Sample $B_s$ samples of $X$ from $\mathcal{D}_{\text{train}}$: $X_l$
10:             Generate knockoff $\tilde{X}_l = g_\theta(X_l)$
11:             Calculate $\mathcal{L}_{\text{SL}}(X_l, \tilde{X}_l, \{S_{\omega_i}\}_{i=1}^K)$ and $\mathcal{L}_{\text{DRL}}(X_l, \tilde{X}_l)$
12:             $\theta \leftarrow \theta + \text{opt}_\theta(\mathcal{L}_{\text{SL}}(X_l, \tilde{X}_l, \{S_{\omega_i}\}_{i=1}^K) + \mathcal{L}_{\text{DRL}}(X_l, \tilde{X}_l))$        ▷ Optimize KT
13:             **if** $l \bmod \gamma = 0$ **then**        ▷ Optimize swappers
14:                 $\omega_i \leftarrow \omega_i + \text{opt}_{\omega_i}(-\mathcal{L}_{\text{SL}}(X_l, \tilde{X}_l, \{S_{\omega_i}\}_{i=1}^K)), i = 1, \ldots, K$
15:             **end if**
16:          **end for**
17:          Calculate the validation loss on all data in $\mathcal{D}_{\text{val}}$: $\mathcal{L}_{\text{SL}}^{\text{val}} + \mathcal{L}_{\text{DRL}}^{\text{val}}$
18:          **if** $\mathcal{L}_{\text{SL}}^{\text{val}} + \mathcal{L}_{\text{DRL}}^{\text{val}}$ meets the early stop condition at tolerance $\eta$ **then**        ▷ Early stopping
19:             **break**
20:          **end if**
21:      **end for**
22: **end procedure**

---

## K    EFFECT OF MODEL SIZE

In this section, we consider the effect of the model size of KT to the feature selection performance. Namely, we change the number of hidden dimensions and the number of layers in half and compare their performance to the default DeepDRK. Results for the hidden dimension case are in Table 3, and results for the number of layers are in Table 4. In general, we find that the model is very robust

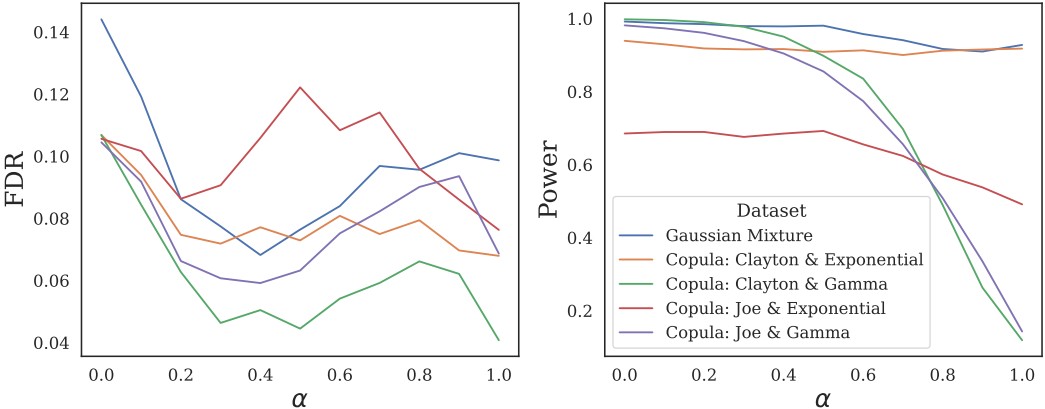

Figure 7: The effect of $\alpha$ for $\tilde{X}_\theta^{\mathrm{DRP}}$ in Eq. (8) on FDR and power. When $\alpha$ is 0, there is only perturbation $X_{\mathrm{rp}}$ without the knockoff. When $\alpha$ is 1, we consider the knockoff generated from the knockoff transformer without any dependency regularization perturbation.

|  | Hidden Dim: 512 | | Hidden Dim: 256 | |
| --- | --- | --- | --- | --- |
| Sample Size | 200 | 2000 | 200 | 2000 |
| Mixture of Gaussians | **0.097/0.851** | **0.075/0.973** | 0.087/0.833 | 0.079/0.971 |
| Copula: Clayton & Exponential | **0.126/0.690** | 0.082/0.875 | 0.131/0.698 | **0.086/0.872** |
| Copula: Clayton & Gamma | **0.080/0.818** | 0.033/0.988 | 0.076/0.816 | **0.032/0.989** |
| Copula: Joe & Exponential | **0.135/0.524** | 0.108/0.665 | 0.158/0.572 | **0.088/0.495** |
| Copula: Joe & Gamma | **0.074/0.774** | **0.064/0.960** | 0.071/0.756 | 0.061/0.955 |

Table 3: The performance comparison between two different hidden dimension setups. The Deep-DRK has 512 hidden dimensions and the reduced model has 256 hidden dimensions. Two sample size setups and five datasets are considered. "·/·" refers to values for "FDR/power". We use "**boldface**" to indicate a better results in this comparison. We consider a result is better when it has higher power and the FDR is controlled at the nominal 0.1, or a lower FDR when FDR is above the 0.1 threshold.

to the change of model size as in many cases in Table 3 and Table 4, the performance of the reduced model is consistent with DeepDRK. This is mostly due to the early stopping criterion that is applied during training, which effectively prevents overfitting. However, there are cases with "Copula: Joe & Exponential" and "Copula: Joe & Gamma" datasets, where the drop in performance can be observed with the reduced models. As a result, we recommand start with the default DeepDRK configuration when applying the framework to other datasets, and change the model size if there is burden in the computational hardware.

## L    ADDITIONAL RESULTS

In this section, we include all results that are deferred from the main paper.

### L.1    FULL SYNTHETIC CASE WITH SAMPLE SIZE 500

Identical to the setup in Section 4.3, we consider the data of sample size 500. The results are included in Figure 8. We can clearly observe the similar behaviour across different models as in the cases with 200 and 2000 sample sizes. For the two different setups, the DeepDRK maintain the lowest FDR among all cases for the two different setups (i.e. Figure 8a and 8b), while keeping a relatively high power, compared to other models.

| | # Layers: 8 | | # Layers: 4 | |
|---|---|---|---|---|
| Sample Size | 200 | 2000 | 200 | 2000 |
| Mixture of Gaussians | **0.097/0.851** | 0.075/0.973 | 0.090/0.828 | **0.075/0.976** |
| Copula: Clayton & Exponential | **0.126/0.690** | 0.082/0.875 | 0.133/0.716 | **0.090/0.888** |
| Copula: Clayton & Gamma | **0.080/0.818** | **0.033/0.988** | 0.084/0.814 | 0.032/0.987 |
| Copula: Joe & Exponential | **0.135/0.524** | **0.108/0.665** | 0.159/0.578 | 0.119/0.665 |
| Copula: Joe & Gamma | **0.074/0.774** | **0.064/0.960** | 0.084/0.674 | 0.145/0.744 |

Table 4: The performance comparison between two different layer setups. The original DeepDRK has 8 layers (e.g. attention blocks) and the reduced model has 4 layers. Two sample size setups and five datasets are considered. "·/·" refers to values for "FDR/power". We use "**boldface**" to indicate a better results in this comparison. We consider a result is better when it has higher power and the FDR is controlled at the nominal 0.1, or a lower FDR when FDR is above the 0.1 threshold.

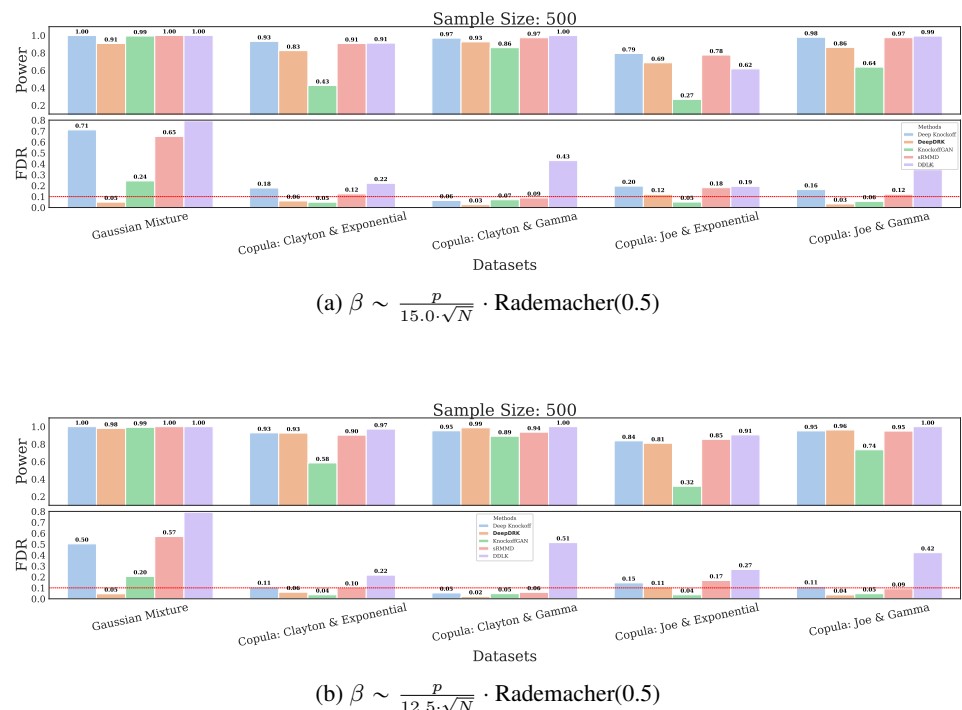

(a) $\beta \sim \frac{p}{15.0 \cdot \sqrt{N}} \cdot \text{Rademacher}(0.5)$

(b) $\beta \sim \frac{p}{12.5 \cdot \sqrt{N}} \cdot \text{Rademacher}(0.5)$

Figure 8: Synthetic datasets with sample size 500. Two different distributions for $\beta$ are considered.

## L.2 SWAP PROPERTY MEASUREMENT

We use different metrics to empirically evaluate the swap property on the generated knockoff $\tilde{X}$ and the original data $X$ (i.e. Eq. (1)). In this paper, three metrics are considered: mean discrepancy distance with linear kernel, or "MMD(Linear)" for short; sliced wasserstein 1 distance (SWD1); and sliced wasserstein 2 distance (SWD2). We measure the sample level distance with the three metrics between the joint vector that concatenates $X$ and $\tilde{X}$ (e.g. $(X, \tilde{X})$) and the joint vector after randomly swapping the entries (e.g. $(X, \tilde{X})_{\text{swap}(B)}$). To avoid repetition, please refer to section 2.1 and Eq. (1) for the definition of notation. Empirically, it is time consuming to evaluate all subsets $B$ of the index set $[p]$. As a result, we alternatively define a swap ratio $r_s \in \{0.1, 0.3, 0.5, 0.7, 0.9\}$. The swap ratio controls the amount of uniformly sampled indices (i.e. the cardinality $|B| = r_s \cdot p$) in a subset of $[p]$. For any $X$ and $\tilde{X}$ from the same experiment, 5 different subsets $B$ are formed according to 5 different swap ratios. And we report average value over the swap ratio to represent the empirical measurement on the swap property. Results can be found in Table 5.

Clearly, compared to other models, the proposed DeepDRK achieves the smallest values in almost every entry across the three metrics and the first 4 datasets (i.e. J+G, C+G, C+E and J+E in Table 5). This explains why DeepDRK has lower FDRs as DeepDRK maintains the swap property relatively better than the benchmarking models (see results in Figure 2 and 3). Similarly, we observe that KnockoffGAN also achieves relatively small values, which leads to well-controlled FDRs compared to other benchmarking models. Overall, this verifies the argument in Candes et al. (2018) that the swap property is important in guaranteeing FDR during feature selection.

Nevertheless, we also identify a discrepancy, which is the case with mixture of Gaussian distributions. The proposed DeepDRK achieves the best performance in FDR control and power (see results in Figure 2 and 3), yet its swap property measured on the three proposed metrics in Table 5 is not the lowest. Despite this counter-intuitive observation, we want to highlight that it does not conflict with the argument in Candes et al. (2018). Rather, it supports our statement that the low reconstructability and the swap property cannot be achieved at the sample level (e.g. the free lunch dilemma in practice). Low measurements can potentially overfit the knockoff to be close to the original data $X$, resulting in high FDR. After all, the swap property is not the only factor that decides FDR and power during feature selection.

|  | Dataset | MMD(Linear) | SWD1 | SWD2 |
|---|---|---|---|---|
| DDLK | J+G | 2.68 | 0.18 | 0.08 |
|  | C+G | 2.49 | 0.18 | 0.07 |
|  | C+E | 1.80 | 0.15 | 0.05 |
|  | J+E | 2.29 | 0.15 | 0.06 |
|  | MG | 306.49 | 2.15 | 9.99 |
| KnockoffGAN | J+G | 7.01 | 0.15 | 0.08 |
|  | C+G | 5.11 | 0.16 | 0.04 |
|  | C+E | 0.52 | 0.06 | 0.01 |
|  | J+E | 1.24 | 0.08 | 0.02 |
|  | MG | 1.08 | 3.28 | 26.09 |
| Deep Knockoff | J+G | 13.09 | 0.21 | 0.11 |
|  | C+G | 19.04 | 0.27 | 0.14 |
|  | C+E | 6.65 | 0.18 | 0.07 |
|  | J+E | 6.78 | 0.19 | 0.13 |
|  | MG | 2770.00 | 8.98 | 196.54 |
| **DeepDRK (Ours)** | J+G | 0.47 | 0.13 | 0.06 |
|  | C+G | 0.71 | 0.14 | 0.05 |
|  | C+E | 0.23 | 0.09 | 0.02 |
|  | J+E | 0.14 | 0.10 | 0.04 |
|  | MG | 380 | 6.13 | 84.37 |
| sRMMD | J+G | 130.02 | 0.68 | 0.71 |
|  | C+G | 175.74 | 0.73 | 0.96 |
|  | C+E | 49.09 | 0.41 | 0.35 |
|  | J+E | 33.24 | 0.35 | 0.29 |
|  | MG | 3040 | 8.51 | 142.99 |

Table 5: Evaluation on the swap property. This table empirically measures the swap property by three different metrics (i.e. MMD(Linear), SWD1 and SWD2). The evaluation considers all benchmarking models and all datasets in the synthetic dataset setup. For space consideration, we use abbreviations to indicate the name of the datasets. The full name can be found in Table 6.

### L.3 ABLATION STUDY

In this section, we perform ablation study on different terms introduced in Section 3.1.1, to show the necessity of designing these terms during the optimization for knockoffs. We consider the fully synthetic setup described in Section 4.3, and consider 200 and 2000 different sample sizes. The distribution for $\beta$ is $\frac{p}{15 \cdot \sqrt{N}} \cdot$ Rademacher(0.5), which is the harder case. Overall, we consider the

| Abbreviation | Full Name |
|---|---|
| J+G | Copula: Joe & Gamma |
| C+G | Copula: Clayton & Gamma |
| C+E | Copula: Clayton & Exponential |
| J+E | Copula: Joe & Exponential |
| MG | Mixture of Gaussians |

Table 6: The abbreviation table for the datasets.

| | w/ SWC (DeepDRK) | | w/o SWC | |
|---|---|---|---|---|
| Sample Size | 200 | 2000 | 200 | 2000 |
| Mixture of Gaussians | **0.097/0.851** | **0.075/0.973** | 0.476/0.927 | 0.100/0.973 |
| Copula: Clayton & Exponential | 0.126/0.690 | **0.082/0.875** | **0.121/0.642** | 0.081/0.872 |
| Copula: Clayton & Gamma | **0.080/0.818** | **0.033/0.988** | 0.072/0.806 | 0.052/0.875 |
| Copula: Joe & Exponential | 0.135/0.524 | **0.108/0.665** | **0.132/0.352** | 0.108/0.551 |
| Copula: Joe & Gamma | **0.074/0.774** | **0.064/0.960** | 0.069/0.756 | 0.068/0.813 |

Table 7: The ablation study comparing DeepDRK w/ and w/o the dependency regularization SWC term. Two sample size setups and five datasets are considered. "·/·" refers to values for "FDR/power". We use "**boldface**" to indicate a better results in this comparison. We consider a result is better when it has higher power and the FDR is controlled at the nominal 0.1, or a lower FDR when FDR is above the 0.1 threshold.

following terms and provide results in the associated tables: 1. SWC (Table 7); 2. REx (Table 8); 3. the number of swappers $K$ (Table 9); 4. $\mathcal{L}_{\text{swapper}}$ (Table 10); 5. $\mathcal{L}_{\text{ED}}$ (Table 11). For each considered term, we conduct experiments with that term removed or a change of term value, and compare the results with the default DeepDRK model. Five synthetic datasets are considered as before and we report "FDR/power" values for each setup. In the tables, we use "**boldface**" to indicate a better results in this comparison. We follow two rules to decide a better result. First, a result is better when it has higher power and the FDR is controlled at the nominal 0.1. This is because the knockoff framework theoretically guarantees the FDR. A well-controlled FDR indicates a good generated knockoff. Given the FDR is controlled, a better power reflects a low reconstructability. The second rule compares FDRs when they are above the 0.1 threshold and choose the result with a lower FDR as controlling the FDR is the goal of the knockoff framework.

| | w/ REx (DeepDRK) | | w/o REx | |
|---|---|---|---|---|
| Sample Size | 200 | 2000 | 200 | 2000 |
| Mixture of Gaussians | **0.097/0.851** | **0.075/0.973** | 0.094/0.831 | 0.112/0.970 |
| Copula: Clayton & Exponential | **0.126/0.690** | **0.082/0.875** | 0.133/0.688 | 0.084/0.872 |
| Copula: Clayton & Gamma | **0.080/0.818** | **0.033/0.988** | 0.074/0.810 | 0.037/0.985 |
| Copula: Joe & Exponential | **0.135/0.524** | **0.108/0.665** | 0.173/0.627 | 0.112/0.646 |
| Copula: Joe & Gamma | **0.074/0.774** | **0.064/0.960** | 0.073/0.772 | 0.064/0.954 |

Table 8: The ablation study comparing DeepDRK w/ and w/o the REx term. Two sample size setups and five datasets are considered. "·/·" refers to values for "FDR/power". We use "**boldface**" to indicate a better results in this comparison. We consider a result is better when it has higher power and the FDR is controlled at the nominal 0.1, or a lower FDR when FDR is above the 0.1 threshold.

In Table 7, we notice that for almost all cases equipped with SWC, the powers are higher than the models with SWC removed, at a marginal cost of the increment in FDR. In Table 8, we observe that DeepDRK achieves lower FDRs in general, compared to the case where the REx term is removed during optimization. This verifies the statement in Section 3.1.1 that the necessity of balancing the generated knockoff against different adversarial swap attacks (e.g. due to nonlinearity in optimization). Moreover, we consider reducing the number of swappers $K$ from 2 to 1 to mimic the

| | $K = 2$ (DeepDRK) | | $K = 1$ | |
|---|---|---|---|---|
| Sample Size | 200 | 2000 | 200 | 2000 |
| Mixture of Gaussians | **0.097/0.851** | **0.075/0.973** | 0.093/0.832 | 0.076/0.971 |
| Copula: Clayton & Exponential | **0.126/0.690** | 0.082/0.875 | 0.131/0.684 | **0.073/0.909** |
| Copula: Clayton & Gamma | **0.080/0.818** | **0.033/0.988** | 0.080/0.777 | 0.044/0.898 |
| Copula: Joe & Exponential | **0.135/0.524** | **0.108/0.665** | 0.153/0.497 | 0.122/0.692 |
| Copula: Joe & Gamma | **0.074/0.774** | **0.064/0.960** | 0.074/0.734 | 0.063/0.855 |

Table 9: The ablation study comparing DeepDRK and the model with only one swapper. $K$ indicates the number of swappers in the model. Two sample size setups and five datasets are considered. "·/·" refers to values for "FDR/power". We use "**boldface**" to indicate a better results in this comparison. We consider a result is better when it has higher power and the FDR is controlled at the nominal 0.1, or a lower FDR when FDR is above the 0.1 threshold.

| | w/ $\mathcal{L}_{\text{swapper}}$ (DeepDRK) | | w/o $\mathcal{L}_{\text{swapper}}$ | |
|---|---|---|---|---|
| Sample Size | 200 | 2000 | 200 | 2000 |
| Mixture of Gaussians | **0.097/0.851** | 0.075/0.973 | 0.096/0.828 | **0.074/0.975** |
| Copula: Clayton & Exponential | 0.126/0.690 | **0.082/0.875** | **0.113/0.655** | 0.083/0.875 |
| Copula: Clayton & Gamma | **0.080/0.818** | **0.033/0.988** | 0.078/0.801 | 0.046/0.917 |
| Copula: Joe & Exponential | **0.135/0.524** | **0.108/0.665** | 0.136/0.469 | 0.121/0.686 |
| Copula: Joe & Gamma | **0.074/0.774** | **0.064/0.960** | 0.072/0.762 | 0.059/0.940 |

Table 10: The ablation study comparing DeepDRK w/ and w/o the cosine similarity term for the swappers: $\mathcal{L}_{\text{swapper}}$. Two sample size setups and five datasets are considered. "·/·" refers to values for "FDR/power". We use "**boldface**" to indicate a better results in this comparison. We consider a result is better when it has higher power and the FDR is controlled at the nominal 0.1, or a lower FDR when FDR is above the 0.1 threshold.

swapper setup in DDLK (Sudarshan et al., 2020). The consistent decrement in FDR (see Table 9) indicate that the multi-swapper setup, as introduced in DeepDRK, works properly in guaranteeing a better swap property. In Table 10, we additionally compare DeepDRK with the model that removes the $\mathcal{L}_{\text{swapper}}$ term, leaving no constraints on the weights in different swappers. This means the two swappers can have the same adversarial swap attack. We can clearly observe the slight decrement in power and increment in FDR in the latter case in this ablation study, indicating the usefulness in forcing different swappers to perform differently during adversarial attacks (e.g. different weights).

| | w/ $\mathcal{L}_{\text{ED}}$ (DeepDRK) | | w/o $\mathcal{L}_{\text{ED}}$ | |
|---|---|---|---|---|
| Sample Size | 200 | 2000 | 200 | 2000 |
| Mixture of Gaussians | **0.097/0.851** | **0.075/0.973** | 0.100/0.816 | 0.077/0.971 |
| Copula: Clayton & Exponential | 0.126/0.690 | 0.082/0.875 | **0.122/0.688** | **0.086/0.880** |
| Copula: Clayton & Gamma | **0.080/0.818** | **0.033/0.988** | 0.071/0.802 | 0.037/0.986 |
| Copula: Joe & Exponential | **0.135/0.524** | 0.108/0.665 | 0.160/0.579 | **0.106/0.666** |
| Copula: Joe & Gamma | 0.074/0.774 | **0.064/0.960** | **0.082/0.778** | 0.064/0.945 |

Table 11: The ablation study comparing DeepDRK w/ and w/o the entry-wise decorrelation term: $\mathcal{L}_{\text{ED}}$. Two sample size setups and five datasets are considered. "·/·" refers to values for "FDR/power". We use "**boldface**" to indicate a better results in this comparison. We consider a result is better when it has higher power and the FDR is controlled at the nominal 0.1, or a lower FDR when FDR is above the 0.1 threshold.

Overall, we verify that all terms are necessary components to achieve higher powers and controlled FDRs through this ablation study.

Besides the above terms, we further consider the $\mathcal{L}_{\text{ED}}$ term in an ablation study. Results can be found in Table 11. This term is primarily used for the purpose of stabilizing the training. We find that the model without this term would fail 2 times out of 10 trials with some datasets. As shown in Table 11, there is only minimal change of performance when $\mathcal{L}_{\text{ED}}$ is removed compared to DeepDRK that equips with this term since most values differ in only the third decimal point. Therefore, we recommend to remove this term first when applying DeepDRK on a dataset for the purpose of reducing the number of hyperparameters and improving the training speed. If the optimization fails on the given dataset, one can add this term back.

## M    MODEL RUNTIME

We consider evaluating and comparing model training runtime in Table 12 with the (2000, 100) setup, as it is common in existing literature. Although DeepDRK is not the fastest among the compared models, the time cost—7.35 minutes—is still short, especially when the performance is taken into account.

| DeepDRK | Deep Knockoff | sRMMD | KnockoffGAN | DDLK |
|---------|---------------|-------|-------------|------|
| 7.35 min | 1.08 min | 6.38 min | 10.52 min | 53.63 min |

Table 12: Average time cost of training models with the $n$ and $p$ setup: (2000, 100). The values are considered with batch size 64 and training for 100 epochs.

## N    PREPARATION OF THE RNA DATA

We first normalize the raw data $X$ to value range $[0, 1]$ and then standardize it to have zero mean and unit variance. $Y$ is synthesized according to $X$. We consider two different ways of synthesizing $Y$. The first is identical to the previous setup in the full synthetic case with $Y \sim \mathcal{N}(X^T\beta, 1)$ and $\beta \sim \frac{p}{12.5 \cdot \sqrt{N}} \cdot \text{Rademacher}(0.5)$. For the second, the response $Y$ is generated following the expression:

$$k \in [m/4]$$
$$\varphi_k^{(1)}, \varphi_k^{(2)} \sim \mathcal{N}(1, 1)$$
$$\varphi_k^{(3)}, \varphi_k^{(4)}, \varphi_k^{(5)} \sim \mathcal{N}(2, 1) \tag{25}$$
$$Y \mid X = \epsilon + \sum_{k=1}^{m/4} \varphi_k^{(1)} X_{4k-3} + \varphi_k^{(3)} X_{4k-2} + \varphi_k^{(4)} \tanh\left(\varphi_k^{(2)} X_{4k-1} + \varphi_k^{(5)} X_{4k}\right),$$

where $\epsilon$ follows the standard normal distribution and the 20 covariates are sampled uniformly.

## O    REAL CASE STUDY

Here we provide the supplementary information for the experiments described in Section 4.5. In Table 13, we provide all the 47 referenced metabolites based on our comprehensive literature review. In Table 14, we provide the list of identified metabolites by each of the considered models. This table corresponds to Table 1 in the main paper that only include metabolites counts due to space limitation.

| Reference Type | Metabolite | Source | Meatbolite | Source |
|---|---|---|---|---|
| PubChem | palmitate | CID: 985 | taurocholate | CID: 6675 |
| | cholate | CID: 221493 | p-hydroxyphenylacetate | CID: 127 |
| | linoleate | CID: 5280450 | deoxycholate | CID: 222528 |
| | taurochenodeoxycholate | CID: 387316 | | |
| Publications | 12.13-diHOME | Bin Masud et al. (2021) | dodecanedioate | Bin Masud et al. (2021) |
| | arachidonate | Bin Masud et al. (2021) | eicosatrienoate | Bin Masud et al. (2021); Bauset et al. (2021) |
| | eicosadienoate | Bin Masud et al. (2021) | docosapentaenoate | Bin Masud et al. (2021); Bauset et al. (2021) |
| | taurolithocholate | Bin Masud et al. (2021) | salicylate | Bin Masud et al. (2021) |
| | saccharin | Bin Masud et al. (2021) | 1.2.3.4-tetrahydro-beta-carboline-1.3-dicarboxylate | Bin Masud et al. (2021) |
| | oleate | Bauset et al. (2021) | arachidate | Bauset et al. (2021) |
| | glycocholate | Bauset et al. (2021) | chenodeoxycholate | Bauset et al. (2021) |
| | phenyllactate | Masud et al. (2021); Lavelle & Sokol (2020) | glycolithocholate | Bauset et al. (2021) |
| | urobilin | Masud et al. (2021); Qin (2012) | caproate | Masud et al. (2021); Lee et al. (2017) |
| | hydrocinnamate | Masud et al. (2021); Koon (2022) | myristate | Masud et al. (2021); Fretland et al. (1990) |
| | adrenate | Masud et al. (2021); Lloyd-Price et al. (2019) | olmesartan | Masud et al. (2021); Saber et al. (2019) |
| | tetradecanedioate | Suhre et al. (2011); Mehta et al. (2022) | hexadecanedioate | Suhre et al. (2011); Mehta et al. (2022) |
| | oxypurinol | Blaker et al. (2013) | porphobilinogen | Minderhoud et al. (2007) |
| | caprate | Soderholm et al. (1998); Söderholm et al. (2002) | undecanedionate | Lee et al. (2017); Uko et al. (2012) |
| | stearate | Ananthakrishnan et al. (2017); Bauset et al. (2021) | oleanate | Nuzzo et al. (2021) |
| | glycochenodeoxycholate | Scoville et al. (2018) | sebacate | Lee et al. (2017) |
| | nervonic acid | Uchiyama et al. (2013) | lithocholate | Bauset et al. (2021) |
| Preprints | alpha-muricholate | Narasimhan et al. (2020) | tauro-alpha-muricholate/tauro-beta-muricholate | Narasimhan et al. (2020) |
| | 17-methylstearate | Narasimhan et al. (2020) | myristoleate | Narasimhan et al. (2020) |
| | taurodeoxycholate | Narasimhan et al. (2020) | ketodeoxycholate | Narasimhan et al. (2020) |

Table 13: IBD-associated metabolites that have evidence in the literature. This table includes all 47 referenced metabolites for the IBD case study. Each metabolite is supported by one of the three evidence sources: PubChem, peer-reviewed publications or preprints. For PubChem case, we report the PubChem reference ID (CID), and for the other two cases we report the publication references.

| Metabolite | **DeepDRK** | Deep Knockoff | sRMMD | KnockoffGAN | DDLK |
|---|---|---|---|---|---|
| 12.13-diHOME | * | | | | * |
| 9.10-diHOME | | | | | |
| caproate | * | * | * | | * |
| hydrocinnamate | * | | | | |
| mandelate | * | | | | |
| 3-hydroxyoctanoate | * | | | | |
| caprate | | | | | |
| indoleacetate | | | | | * |
| 3-hydroxydecanoate | | | | | |
| dodecanoate | * | | | * | |
| undecanedionate | * | | | * | |
| myristoleate | * | | | | |
| myristate | * | | | | |
| dodecanedioate | * | | | * | |
| pentadecanoate | | | | | |
| hydroxymyristate | | | | | |
| palmitoleate | | | | | |
| palmitate | * | | | * | |
| tetradecanedioate | * | * | | | |
| 10-heptadecenoate | | | | | |
| 2-hydroxyhexadecanoate | | | | | |
| alpha-linolenate | | | | | * |
| linoleate | * | | | | |
| oleate | * | | | | |
| stearate | * | | | | * |
| hexadecanedioate | * | * | | * | * |
| 10-nonadecenoate | * | | | | |
| nonadecanoate | | | | | |
| 17-methylstearate | * | * | | | * |
| eicosapentaenoate | * | * | | | * |
| arachidonate | * | * | | * | * |
| eicosatrienoate | * | * | | | * |
| eicosadienoate | * | * | * | | * |
| eicosenoate | * | | | | |
| arachidate | * | | | * | |
| phytanate | | | | | |
| docosahexaenoate | | * | | | * |
| docosapentaenoate | * | * | | * | * |
| adrenate | * | * | * | * | * |
| 13-docosenoate | * | | | | |
| eicosanedioate | * | * | | | |
| oleanate | | | | | |
| masilinate | | | | | |
| lithocholate | | | | | |
| chenodeoxycholate | * | | | | |
| deoxycholate | * | | | * | * |
| hyodeoxycholate/ursodeoxycholate | | | | | |
| ketodeoxycholate | * | | | | |
| alpha-muricholate | * | | | | |
| cholate | | * | | | |
| glycolithocholate | * | | | | |
| glycochenodeoxycholate | * | | | | |
| glycodeoxycholate | | | | | |
| glycoursodeoxycholate | * | | | | |
| glycocholate | | | | | |
| taurolithocholate | | | | | * |
| taurochenodeoxycholate | * | | | | |
| taurodeoxycholate | * | | | | |
| taurohyodeoxycholate/tauroursodeoxycholate | * | | | | |
| tauro-alpha-muricholate/tauro-beta-muricholate | * | * | | | * |
| taurocholate | * | | | | |
| salicylate | * | * | * | * | |
| saccharin | * | | | * | |
| azelate | | | | | * |
| sebacate | | | | | * |
| carboxyibuprofen | | | | | |
| olmesartan | | | | | |
| 1.2.3.4-tetrahydro-beta-carboline-1.3-dicarboxylate | * | | | | |
| 4-hydroxystyrene | * | * | | * | * |
| acetytyrosine | | | | | |
| alpha-CEHC | * | | | | |
| carnosol | * | | | | * |
| oxypurinol | | | | | |
| palmitoylethanolamide | * | | | | |
| phenyllactate | * | * | * | | * |
| p-hydroxyphenylacetate | | * | | | * |
| porphobilinogen | * | | | | |
| urobilin | * | * | | * | * |
| nervonic acid | * | | | | |
| oxymetazoline | * | * | | | * |

Table 14: A list of identified metabolites out of the total 80. "∗" indicates the important metabolite marked by the corresponding algorithms.

