# OpenReview forum: "DeepDRK: Deep Dependency Regularized Knockoff for Feature Selection"
_ICLR.cc/2024/Conference — Submitted to ICLR 2024_

### Official Review · Reviewer_CLhf · 2023-10-30

**Soundness:** 3 good
**Presentation:** 3 good
**Contribution:** 3 good
**Rating:** 5
**Confidence:** 2

**Summary:**

The authors develop a distribution-free deep learning method for knockoff generation which strikes a balance between FDR and power, called “Deep Dependency Regularized Knockoff” (DeepDRK). In DeepDRK, a “multi-swapper” adversarial training procedure is proposed to enforce the swap property, while a sliced-Wasserstein-based dependency regularization (together with a novel perturbation technique) is introduced to reduce reconstructability. Experiments on real, synthetic, and semi-synthetic datasets are carried out to show the good performance.

**Strengths:**

1) Proposed a distribution-free deep learning method for knockoff generation which strikes a balance between FDR and power.
2) A DeepDRK pipeline is provided to increase readability.
3) A number of experimental results are provided on simulated, semi-simulated and real datasets to illustrate the performance of the proposed method.

**Weaknesses:**

1) Though it enjoys theoretical result that a no free lunch situation for selection power when there is exact reconstructability in Appendix B, it seem that there is no theoretical guarantees on the power or explanations for that how the sliced-Wasserstein-based dependency regularization together with a novel perturbation technique introduced to reduce reconstructability can promote selection power.
2) It is not clear that how to enforce the swap property by the “multi-swapper” adversarial training procedure.
3) The motivation behind feature selection is high-dimensional data settings, which in my understanding means that the number of features is larger than the number of examples in the dataset. However, none of the simulated experiments include such scenario.

Examples of writing problems:
-“Similar observations can be found in Figure 4.” seems to be “Similar observations can be found in Figure 5.” in the paragraph “Results” of section 4.4.
-“Among them, model-specific ones such as AEknockoff (Liu & Zheng, 2018) Hidden Markov Model (HMM), knockoff (Sesia et al., 2017)” seems to be “Among them, model-specific ones such as AEknockoff (Liu & Zheng, 2018), Hidden Markov Model (HMM) knockoff (Sesia et al., 2017)” in the first paragraph of section 2.2.

**Questions:**

(1) More explanations for the proposed method striking a balance between FDR and power.
(2)The diagram of DeepDRK pipeline and code library are given, but the algorithm for training objective (4) is not provided.

---

> ### Author Response · Authors · 2023-11-19
> **Authors' Response (part 1)**
>
> Thank you for your insightful and constructive feedback on our manuscript. We appreciate the time and effort you invested in reviewing our work. We have carefully considered your comments and have made the following revisions to address the concerns raised above:
>
> ## Weaknesses
>
> - Though it enjoys theoretical result that a no free lunch situation for selection power when there is exact reconstructability in Appendix B, it seem that there is no theoretical guarantees on the power or explanations for that how the sliced-Wasserstein-based dependency regularization together with a novel perturbation technique introduced to reduce reconstructability can promote selection power.
>
> **ANSWER**: To the best of our knowledge, theoretical analysis on FDR and power are mostly restricted to parametric designs. Without assuming the knowledge of the feature distribution (i.e. the distribution of $X$), how the SWC dependency regularization and the perturbation affect FDR (Type I error) and power (Type II error) is still an open problem, and we leave it for future research. Intuitively, adding independent components leads to less collinearity and thus improves power, at the cost that the swap property will be violated. Empirically, we observed that by enforcing swap property, the generated knockoff copies are often too close to the original design matrix, resulting in high collinearity so that although Type I error might be controlled, Type II error can be large. In such cases, we aim to sacrifice some FDR control for higher power in return. Experiments show that the SWC dependency regularization and the perturbation will result in slightly larger FDR (still relatively small), and much improved power (see Table 7 and Figure 7).
>
> - It is not clear that how to enforce the swap property by the “multi-swapper” adversarial training procedure.
>
> **ANSWER**:  Thank you for pointing out this issue. We have added the description of the design of swappers and how they assist the knockoff transformer in achieving the swap property in the second paragraph of Appendix C. To repeat the part that involves the optimization: To generate the subset $B$, we consider drawing samples of $b_j$ from the corresponding $j$-th Gumbel-softmax random variable, and the subset $B$ is defined as $\\{j\in [p] ; b\_j=1\\}$. During optimization, we maximize Eq. (4) w.r.t. to the weights $\omega_i$ of the swapper $S_{\omega_i}$ such that the sampled indices, with which the swap is applied, lead to a higher $\text{SWD}$ in the objective (Eq. (4)). Minimizing this objective w.r.t. $\tilde{X}_\theta$ requires the knockoff to fight against the adversarial swaps. Therefore, the swap property is enforced. Compared to DDLK, the proposed DeepDRK utilizes multiple independent swappers.
>
> We suggest checking Appendix C for the full description of how swappers are defined and used during this optimization.
>
> - The motivation behind feature selection is high-dimensional data settings, which in my understanding means that the number of features is larger than the number of examples in the dataset. However, none of the simulated experiments include such scenario.
>
> **ANSWER**: Thank you for pointing out this part. To the best of our knowledge, existing literature `[1-5]`, both theoretical or methodological, concerns the classical linear regression setting when $n>p$ (or $n>2p$ in the knockoff framework as the knockoff introduces additional $p$ variables to the independent variables) . Extensions to the high dimensional setting are left to be unveiled.
>
> We also want to mention that the knockoff generation part (e.g. the main objective of DeepDRK), as discussed in `[1]`, can handle high dimensional data as there is no restriction on the generative model part in terms of the dimensionality. Hence, it is not a conflict of the statement in a high dimensional setting to the existing literature.
>
>
> `[1]` Emmanuel Candes, Yingying Fan, Lucas Janson, and Jinchi Lv. Panning for gold:‘Model-X’ knockoffs for high dimensional controlled variable selection. Journal of the Royal Statistical Society: Series B (Statistical Methodology), 80(3):551–577, 2018.
>
> `[2]` Yaniv Romano, Matteo Sesia, and Emmanuel Candes. Deep knockoffs. Journal of the American Statistical Association, 115(532):1861–1872, 2020.
>
> `[3]` James Jordon, Jinsung Yoon, and Mihaela van der Schaar. Knockoffgan: Generating knockoffs for feature selection using generative adversarial networks. In International Conference on Learning Representations, 2018.
>
> `[4]` Shoaib Bin Masud, Matthew Werenski, James M Murphy, and Shuchin Aeron. Multivariate rank via entropic optimal transport: sample efficiency and generative modeling. arXiv preprint arXiv:2111.00043, 2021.
>
> `[5]` Mukund Sudarshan, Wesley Tansey, and Rajesh Ranganath. Deep direct likelihood knockoffs. Advances in neural information processing systems, 33:5036–5046, 2020.

---

> > ### Author Response · Authors · 2023-11-19
> > **Authors' Response (part 2, continued)**
> >
> > - Examples of writing problems: “Similar observations can be found in Figure 4.” seems to be “Similar observations can be found in Figure 5.” in the paragraph “Results” of section 4.4. -“Among them, model-specific ones such as AEknockoff (Liu & Zheng, 2018) Hidden Markov Model (HMM), knockoff (Sesia et al., 2017)” seems to be “Among them, model-specific ones such as AEknockoff (Liu & Zheng, 2018), Hidden Markov Model (HMM) knockoff (Sesia et al., 2017)” in the first paragraph of section 2.2.
> >
> > **ANSWER**: Thank you for pointing out these issues. We have corrected the typos in the revised manuscript.
> >
> >
> > ## Questions
> >
> > - More explanations for the proposed method striking a balance between FDR and power.
> >
> > **ANSWER**: Thank you for raising this point. In the original paper, we have provided some explanation in Sections 2.3 and 3.1.2 in the original paper. In the revised manuscript, we have added additional two sets of experiments to clarify it. The first experiment is to investigate the effect of $\alpha$ in the $\tilde{X}^{\text{DRP}}\_\theta$ term (Eq. (8)) to the performance of FDR and power, where $\alpha$ controls the portion of the introduction of $X\_{\text{rp}}$, the permuted $X$. Results can be found in Figure 7 in Appendix J of the revised manuscript. We verified that decreasing the value of $\alpha$ increases the power, however, at the cost of increasing the FDR. This is because the permutation inside $\tilde{X}^{\text{DRP}}\_\theta$ reduces reconstructability and destroys swap property simultaneously. Overall, this experiment explains why it is crucial in consideration of the DRP term for feature selection.
> >
> > Another experiment we considered measures the goodness of fit on the swap property, which elaborates how swap property is related to the power and FDR at sample level. Details can be found in Appendix L.2 in the revised manuscript. We also summarize this part in the “Measurement on Swap Property” of the general response above. Additionally, we want to emphasize that the discussion of the balance between FDR and power is already included in the original manuscript: In section 2.3, we discussed why reconstructability can affect power. Whereas in section 3.1.2, we provided explanations on why we introduced Eq. (7) (i.e. the SWC and the $\mathcal{L}\_{\text{ED}}$) for reducing the reconstructability. This trades off the power and FDR.
> >
> > - The diagram of DeepDRK pipeline and code library are given, but the algorithm for training objective (4) is not provided.
> >
> > **ANSWER**: Thank you for pointing this out. We have added the training algorithm (for the objective in Eq. (4)) in Algorithm 1 of Appendix I.

---

> > > ### Author Response · Authors · 2023-11-21
> > > **Look Forward to Your Further Feedback**
> > >
> > > Dear Reviewer CLhf,
> > >
> > > We are grateful for your insightful and constructive comments on our paper. We have addressed your comments in detail with the revised manuscript ([link](https://openreview.net/pdf?id=0SOhDO7xI0)).
> > >
> > > We would like to bring your attention to these updates and eagerly await any additional feedback. Your insights are incredibly valuable and help us refine our paper.
> > >
> > > Best regards,
> > >
> > > The Authors of Paper 4515

---

### Official Review · Reviewer_Zneh · 2023-10-31

**Soundness:** 3 good
**Presentation:** 2 fair
**Contribution:** 2 fair
**Rating:** 6
**Confidence:** 2

**Summary:**

This paper proposes DeepDRK, a new model-X knockoff based methods which adopts a two-stage framework to generate knockoff variables. A ViT (called knockoff transformer in the paper) is trained by minimizing a swap loss plus a dependency regularization loss in the training stage, while its output is further perturb through a row-permuted version of the original covariate to reduce the dependency between knockoffs and original covaraites. Experiments on synthetic, semi-synthetic and real data demonstrates the effectiveness of DeepDRK.

**Strengths:**

1. Writing is good and it is easy to follow
2. The idea of leveraging distribution-free methods while avoiding overfitting is well motivated.
3. Experiment results are impressive.

**Weaknesses:**

1. Ablation study is not very thorough.
- The loss in DeepDRK contains five terms, SWD, REx, cosine similarity w.r.t. swappers, SWC and the entry-wise decorrelation term. The necessity of introducing these five losses is under-explored in the paper.
- The necessity of DRP is unclear. There lacks comparison of $\tilde{X}_{\theta}$ and $\tilde{X}_{\theta}^{DRP}$ in empirical performance.
2. Experiments need further analysis and explanation.
- It is clear that DeepDRK performs better than other baseline methods. But the reason has not been analyzed clearly and adding some intermediate results will be helpful. It is unclear how well the knockoffs generated by DeepDRK following the swap property and avoid overfitting compared to baseline methods.
- The results w.r.t. the Gaussian mixture seems inconsistent with that in the original DDLK paper (DDLK performs the worst in this paper while it performs better than deep knockoffs and knockoffgan in the original paper).

**Questions:**

To avoid overfitting, why introducing a post-training perturbation instead of modifying training strategy like early stopping or tuning hypermeters?

---

> ### Author Response · Authors · 2023-11-19
> **Authors' Response (part 1)**
>
> Thank you for your insightful and constructive feedback on our manuscript. We have carefully considered your comments and have made the following revisions to address the concerns raised above:
>
> ## Weaknesses
> - The loss in DeepDRK contains five terms, SWD, REx, cosine similarity w.r.t. swappers, SWC and the entry-wise decorrelation term. The necessity of introducing these five losses is under-explored in the paper.
>
> **ANSWER**: Thank you for pointing out this issue. We conducted additional experiments for the ablation study section in Appendix L.3, and summarized them in the “Ablation Study Results” section of the general response above. Additionally, we want to point out that the SWD term is designed for imposing the swap property, which serves as a backbone of the model. Therefore, it cannot be removed as this would lead to complete deconstruction of the knockoff learning problem in our case.
>
> - The necessity of DRP is unclear. There lacks comparison of $\tilde{X}\_{\theta}$ and $\tilde{X}\_{\theta}^{DRP}$ in empirical performance.
>
> **ANSWER**: Thank you for pointing out this issue. We added one experiment to investigate the effect of $\alpha$ to the performance of FDR and power, where $\alpha$ controls the portion of the introduction of $X\_{\text{rp}}$, the perturbed $X$. Results can be found in Figure 7 in Appendix J of the revised manuscript. We verified that decreasing the value of $\alpha$ increases the power, however, at the cost of increasing the FDR. This is because the permutation inside $\tilde{X}^{\text{DRP}}\_\theta$ reduces reconstructability and destroys swap property simultaneously. And $\alpha = 1$ refers to the case without DRP (i.e. $\tilde{X}\_\theta$). Based on our result, we suggest choosing the value $\alpha$ between 0.4 and 0.5. And all results reported in our paper are based on the choice of $\alpha$ equals 0.5.
>
> - It is clear that DeepDRK performs better than other baseline methods. But the reason has not been analyzed clearly and adding some intermediate results will be helpful. It is unclear how well the knockoffs generated by DeepDRK following the swap property and avoid overfitting compared to baseline methods.
>
> **ANSWER**: Thank you for pointing out this part. We included a section "Appendix L.2" in the revised manuscript for measuring the swap property using three different metrics: mean discrepancy distance with the linear kernel and sliced Wasserstein 1 & 2 distances. This reveals the relationship between the sample level swap property and the performance in feature selection. We summarize this part in the “Measurement on Swap Property'' of the general response above.
>
> Besides the verification on the swap property, we also clarified that we avoid overfitting via early stopping in the revised manuscript. Details can be found in Appendix I, which is a newly added section that includes the training algorithm of the knockoff transformer. Its effectiveness is proved given the results in Table 3 & 4, compared to other benchmarking models.
>
> - The results w.r.t. the Gaussian mixture seems inconsistent with that in the original DDLK paper (DDLK performs the worst in this paper while it performs better than deep knockoffs and knockoffgan in the original paper).
>
> **ANSWER**: Thank you for pointing out this part. However, we want to point out that the Gaussian mixture experiment considered in our paper is different than that in the original DDLK paper as we considered a lower signal strength (i.e. $\frac{p}{15\cdot \sqrt{n}}\cdot \text{Rademacher(0.5)}$ or $\frac{p}{12.5\cdot \sqrt{n}}\cdot \text{Rademacher(0.5)}$ for the distribution of $\beta$). The original DDLK uses $\frac{p}{1 \cdot \sqrt{n}}\cdot \text{Rademacher(0.5)}$ as the signal strength, which is larger than ours. The reason we chose the lower signal strength cases is because we found all models perform similarly well on the original case. Descriptions and details concerning this experiment setup can be found in the first paragraph of section 4.3.
>
> Additionally, we also want to mention that in the references `[1, 2]` listed below, the authors independently identified the underperformance of DDLK on some Mixture of Gaussian data and on some real datasets. Therefore, our results in this paper are not inconsistent with the current literature.
>
> `[1]` Derek Hansen, Brian Manzo, and Jeffrey Regier. Normalizing flows for knockoff-free controlled feature selection. Advances in Neural Information Processing Systems, 35:16125–16137, 2022.
>
> `[2]` Shoaib Bin Masud, Matthew Werenski, James M Murphy, and Shuchin Aeron. Multivariate
> rank via entropic optimal transport: sample efficiency and generative modeling. arXiv preprint arXiv:2111.00043, 2021.

---

> > ### Author Response · Authors · 2023-11-19
> > **Authors' Response (part 2, continued)**
> >
> > ## Questions
> >
> > -  To avoid overfitting, why introduce a post-training perturbation instead of modifying training strategy like early stopping or tuning hyperparameters?
> >
> > **ANSWER**: Thank you for raising this question. First, sorry for missing its description in the original manuscript. Indeed, we considered applying the early stopping to prevent overfitting. Details can be found in Appendix I, which is a newly added section that includes the training algorithm of the knockoff transformer. Its effectiveness is shown empirically (see results in Table 3 & 4 compared to other benchmarking models). Besides, we want to mention that we have done hyperparameter tuning and presented the hyperparameters setup in Table 2 of Appendix H, which was already included in the original paper. All the experiments are conducted with this single set of hyperparameter configuration. The consistent performance demonstrates that the model is stable given this configuration across different datasets. By offering this configuration, we aim to reduce the time spent searching for new hyperparameter setups. People can load this default DeepDRK model and start from here.
> >
> > Regarding the post-training perturbation. We want to point out that this is not designed to prevent overfitting. Rather, this technique is important for mitigating the competing relationship between the swap property and the reconstructability (see section 3.2 for the discussion) based on our novel discovery. Because the existing algorithms focus on learning knockoffs via distributional losses, which, by nature, cannot solve the conflicts between the swap property and the reconstructability. Overly addressing the swap property leads to a high reconstructability for lowering the power. We provide this DRP as an efficient remedy to reduce the reconstructability via sample-level perturbation. This, as shown in all results for DeepDRK, works empirically very well compared to the benchmarking models. Pointing out this conflict is one crucial discovery and contribution in this paper as this would prevent other researchers on developing algorithms for knockoff generation from this pitfall.

---

> > > ### Author Response · Authors · 2023-11-21
> > > **Look Forward to Your Further Feedback**
> > >
> > > Dear Reviewer Zneh,
> > >
> > > We are grateful for your insightful and constructive comments on our paper. We have addressed your comments in detail with the revised manuscript ([link](https://openreview.net/pdf?id=0SOhDO7xI0)).
> > >
> > > We would like to bring your attention to these updates and eagerly await any additional feedback. Your insights are incredibly valuable and help us refine our paper.
> > >
> > > Best regards,
> > >
> > > The Authors of Paper 4515

---

### Official Review · Reviewer_cU77 · 2023-11-02

**Soundness:** 3 good
**Presentation:** 3 good
**Contribution:** 3 good
**Rating:** 6
**Confidence:** 3

**Summary:**

The paper proposed “Deep Dependency Regularized Knockoff (DeepDRK)”, a distribution-free deep learning method that strikes a balance between FDR and power. It leverages transformer architecture and several loss functions for training to generate Knockoff.

**Strengths:**

1. It introduces knockoff Transformer to generate knockoff with different regularizations. And it uses multi-swappers to ensure to swap property of generated knockoff.
2. Experimental results show the effectiveness of the proposed method compared to other deep model based knockoff methods.

**Weaknesses:**

1. Some arguments of the proposed method is not validated with corresponding experimental results. For example, “multi-swapper” is used to better achieve swap property. But there are no experiments to justify the how swap property changes when changing from single swapper to multi-swapper. I think authors should also introduce how to empirically measure the swap property. Since the proposed method relies on regularization to enforce the swap property, which is not guaranteed by design.
2. The proposed method uses many regularization terms. Some of the regularization terms have ablation studies, but others are not. For example, L_swapper and L_ED are not included. The effect of $\alpha$ in Eq.~(9) is also not investigated. Moreover, there are four hyperparameters require tuning, making the proposed method hard to tune.
3. The regularization terms largely come from existing papers; I think authors should better justify what is their contribution on top of existing papers.

**Questions:**

Since most dataset is not very large, but the model size is quite large. Did authors try to change the model size to see how it impacts the performance? Maybe the model could be smaller.

---

> ### Author Response · Authors · 2023-11-19
> **Authors' Response**
>
> Thank you for your insightful and constructive feedback on our manuscript. We appreciate the time and effort you invested in reviewing our work. We have carefully considered your comments and have made the following revisions to address the concerns raised above:
>
> ## Weaknesses
> - Some arguments of the proposed method are not validated with corresponding experimental results. For example, “multi-swapper” is used to better achieve swap properties. But there are no experiments to justify how swap property changes when changing from single swapper to multi-swapper. I think authors should also introduce how to empirically measure the swap property. Since the proposed method relies on regularization to enforce the swap property, which is not guaranteed by design.
>
> **ANSWER**: Thank you for pointing out. We included the ablation study on all the regularization terms in the Appendix L.3 of the revised manuscript. Please also see the “Ablation Study Results” section in the general response above for the complete description of the revised ablation study. We also included the section of Appendix L.2. for measuring the swap property. The summary of this experiment can be found in the “Measurement on Swap Property” section of the general response above.
>
>
> - The proposed method uses many regularization terms. Some of the regularization terms have ablation studies, but others are not. For example, L_swapper and L_ED are not included. The effect of $\alpha$ in Eq.~(9) is also not investigated. Moreover, there are four hyperparameters that require tuning, making the proposed method hard to tune.
>
> **ANSWER**: Thank you for pointing out this. As mentioned previously, we completed additional ablation studies considering more terms proposed in this paper, which includes the study of both $\mathcal{L}\_\text{swapper}$ and $\mathcal{L}\_\text{ED}$. Results can be found in Appendix L.3 in the revised manuscript. And the summary can be found in the "Ablation Study Results" section of the general response above. Based on the ablation study, we verified that $\mathcal{L}\_\text{ED}$ only affects the model’s training stability, rather than its feature selection performance. As a result, we suggest removing this term if no instability issue is observed during training. And the resulting number of tuning hyperparameters during training is identical with KnockoffGAN.
>
> Besides the ablation study, we also investigated the effect of $\alpha$ on the performance of FDR and power. Results can be found in Figure 7 in Appendix J of the revised manuscript. We verified that decreasing the value of $\alpha$ increases the power, however, at the cost of increasing the FDR. This is because the permutation inside $\tilde{X}^{\text{DRP}}\_\theta$ reduces reconstructability and deteriorates swap property simultaneously. We also want to point out that this hyperparameter appears  after training, which can be adjusted without introducing any training burden.
>
> - The regularization terms largely come from existing papers; I think authors should better justify what is their contribution on top of existing papers.
>
> **ANSWER**: Thank you for bringing up this point. We added one paragraph to the last part of section 3.1.1 to summarize how these terms collaboratively contribute to the guarantee of swap property. We want to point out that these terms are first used in knockoff generation for feature generation in this paper, which improves the effectiveness of the generation of knockoffs at sample level. And the utilization of these regularization terms is only one part of the contribution of this paper. We additionally summarized the overall contribution of our paper in the general response above in threefold.
>
> ## Questions
> - Since most dataset is not very large, but the model size is quite large. Did authors try to change the model size to see how it impacts the performance? Maybe the model could be smaller.
>
> **ANSWER**: Thank you for bringing up this point about the model size effect. We prepared new experiments and investigated the effect of the model size (in different hidden dimensions and in different numbers of layers) to the performance in FDR and power. Results can be found in Table 3 & 4 in Appendix K. We observed slightly reduced power when considering smaller models in the number of layers and in the number of hidden dimensions. However, in many cases, since we observed similar performance between DeepDRK (e.g. the current setup) and the reduced models, we think the model size is not a strong influence factor to the feature selection performance. Besides, we also applied early stopping during the model training (stated in Appendix I, a newly added section that includes the training algorithm), which is an effective approach (given the results in Table 3 & 4) to prevent potential overfitting issues. This should also mitigate the effect of the model size to the performance of the model.

---

> > ### Author Response · Authors · 2023-11-21
> > **Look Forward to Your Further Feedback**
> >
> > Dear Reviewer cU77,
> >
> > We are grateful for your insightful and constructive comments on our paper. We have addressed your comments in detail with the revised manuscript ([link](https://openreview.net/pdf?id=0SOhDO7xI0)).
> >
> > We would like to bring your attention to these updates and eagerly await any additional feedback. Your insights are incredibly valuable and help us refine our paper.
> >
> > Best regards
> >
> > The Authors of Paper 4515

---

### Official Review · Reviewer_e4Vy · 2023-11-05

**Soundness:** 3 good
**Presentation:** 3 good
**Contribution:** 3 good
**Rating:** 6
**Confidence:** 2

**Summary:**

This paper investigates the problem of feature selection from the perspective of Model-X knockoff owing to its guarantee of false discovery rate (FDR) control.  Realizing the diminished selection power caused by the swap property that knockoffs need, the authors proposed a  Deep Dependency Regularized Knockoff (DeepDRK), which is a distribution-free deep learning method that strikes a balance between FDR and power.  Experiments on synthetic, semi-synthetic, and real-world data verify the effectiveness of the proposed DeepDRK method.

**Strengths:**

1. This paper is well-written and easy to follow.
2. This paper has a clear motivation for diminished selection power caused by the swap property that knockoffs need.
3. Comprehensive experiments on synthetic, semi-synthetic, and real-world data are conducted, which verify the effectiveness of the proposed DeepDRK method.
4. To me, such a distribution-free deep learning method that strikes a balance between FDR and power is new and novel.

**Weaknesses:**

I don't see any major weakness in this work.

**Questions:**

I have no more questions. I am not an expert in this field, but I feel this paper is good from the perspective of general machine learning.

**Details Of Ethics Concerns:**

No ethics review is needed.

---

> ### Author Response · Authors · 2023-11-19
> **Author's Response**
>
> Thanks a lot for your appreciation of this work. We further improved the manuscript based on reviewers’ comments and suggestions. Please see the general response to all reviewers for details.

---

> > ### Comment · Reviewer_e4Vy · 2023-12-03
> > **Response to Authors**
> >
> > Thanks for the response from the authors. After reading reviews from other reviewers and the author's response, I decided to maintain my score.

---

### Author Response · Authors · 2023-11-19
**General Response (part 1)**

Dear reviewers,

Thank you for your time and the insightful feedback. We took careful considerations and added further experiments and clarifications to the original version. We believe that the quality and the clarity of the manuscript has been improved in this way. In what follows, we first provide a brief summary of the changes to the revised manuscript. We then introduce the ablation studies we added into the paper. A discussion on how DeepDRK maintains the swap property together with further implications. Finally we outline the contribution of this work in general.

## Changes in the Revised Manuscript
The revised paper has the following changes: 1. We included the complete ablation study on all regularization terms introduced in Eq. (4) and demonstrated their desired functionalities (Details can be found in Appendix L.3); 2. We included experiments on the empirical measurement of the swap property (see Appendix L.2). This explains why the proposed DeepDRK outperformed other benchmark models; 3. We included an algorithm table (see Algorithm 1 in Appendix I) to assist readers in better understanding the training process of the knockoff transformer; 4. We evaluated the effect of $\alpha$ in DRP and the model size (e.g. the change of hidden dimension or the number of attention layers) to the model's feature selection performance. Details can be found in Appendix J & K, respectively; 5. We also detailed the construction of the swappers and the process of generating adversarial swaps with these swappers in Appendix C.

## Ablation Study Results
We provide a summary of all ablation studies as the issue is raised by most reviewers. In the earlier version, the ablation study includes the SWC and REx terms. We add experiments for the ablation studies of all the rest regularization terms in Appendix J and L.3 in the revised manuscript. In all, the effect of the following six terms is analyzed:
1. SWC (Table 7)
2. REx (Table 8)
3. The number of swappers $K$ (Table 9)
4. $\mathcal{L}\_\text{swapper}$ (a.k.a. The cosine similarity , Table 10)
5. The perturbation $X\_\text{rp}$ (e.g. $\alpha$ in Eq. (8), Figure 7)
6. $\mathcal{L}\_\text{ED}$.

We demonstrate that 1 - 5 loss terms help to improve knockoff generation quality, while 6 contributes to training stability. More specifically, the SWC term generally provides a higher power with similar FDR compared to the case without it. The REx term, the $\mathcal{L}\_\text{swapper}$ term and multi-swapper (i.e. $K=2$) all contribute to either boosting power when FDR is under control, or decreasing FDR when it is above nominal threshold. The effect of the perturbation is studied by tuning the coefficient $\alpha$ from 0 to 1. We show that for $\alpha$ not being too small, FDR is still under control while power is significantly improved. The sixth term $\mathcal{L}\_\text{ED}$ is primarily for the purpose of training stability. We reported that the DeepDRK model without this term would lead to a 20% failure rate on the training process (i.e. resulting in NAN). When training is successful, the feature selection performance is consistent with or without this term, as shown in Table 11. To reduce the number of hyperparameters, we suggest setting the coefficient of this term $\lambda_4$ to 0 in the first place, and reset it if training fails. In the manuscript, we abbreviate the description of $\mathcal{L}\_\text{ED}$ in the main draft and defer more details to Appendix F. In this way, DeepDRK has 4 hyperparameters (3 in training and 1 post-training), which is comparable to the three-hyperparameter setting in both Deep Knockoff and KnockoffGAN. It is worth noting that although DDLK has only 1 hyperparameter, it requires learning the distribution, which is significantly more complicated compared to DeepDRK.

---

> ### Author Response · Authors · 2023-11-19
> **General Response (part 2, continued)**
>
> ## Measurement on Swap Property
> Equipped with all above losses, in Appendix L.2 we compare the swap property of DeepDRK knockoff and baseline methods using different metrics as swap loss, namely mean discrepancy distance with the linear kernel, and the sliced-Wasserstein distances. The results can be found in Table 5. We observed that, in most cases, DeepDRK achieves the smallest values on the swap loss compared to other benchmarking models. This explains why DeepDRK outperforms the other models on FDR. We also found out that in the case with the mixture of Gaussian data, the swap losses for DeepDRK are not the smallest compared to KnockoffGAN, yet DeepDRK is the only model that controls the FDR and yields relatively high power. This suggests that DeepDRK successfully trades in a small fraction of FDR for considerable power improvement. Supporting evidence can also be found in Figure 7 of Appendix J, where the effect of the perturbation coefficient is studied.
>
> ## Summary of the Contribution
> We provide an outline of the contribution of this work in threefold: First, the competing nature between the swap property and reconstructability in knockoffs are identified, which turns out to be crucial when designing (non-parametric) knockoff algorithms for feature selection. Second, to address the issue, we propose the DeepDRK pipeline for generating knockoff copies with novel dependency regularizations. We use experimental studies to showcase the necessity for each term. Third, we demonstrate the effectiveness of DeepDRK on synthetic, semi-synthetic and real-world datasets. By comparing to benchmarking models, we show that DeepDRK generates high quality knockoffs, such that the feature selection across various datasets achieves high power and low FDR.

---

### Meta-Review · Area_Chair_Rmar · 2023-12-07

**Metareview:**

This work presents a method for performing feature selection in deep learning models called DeepDRK, building on top of Model-X knockoffs (Barber & Candes, 2015).  The authors note that while a variety of methods have been developed to use knockoffs in deep learning models, the methods are not robust to different distributions of data, are hard to train and perform poorly under small numbers of observations.   Thus they proposed a regularized version of knockoffs known as DeepDRK.  DeepDRK uses a vision transformer as the underlying model architecture and is designed to maintain the 'swap property' from Barber & Candes while minimizing reconstructability.  They also develop an adversarial training procedure called 'multi-swapper'.

The reviewers found the paper sound overall, well written and easy to follow.  They seemed to agree that feature selection for deep learning was important and interesting.  The review scores were quite borderline (6, 6, 5, 6) with very low variance.  Unfortunately, no reviewer was willing to champion the paper, and similarly none were strongly opposed to acceptance.  There was a consensus among reviewers that there were insufficient ablations done to justify the various modeling and algorithmic choices proposed in the paper.  Unfortunately, given that no reviewers were willing to champion the paper (some did respond during the response phase that they were unwilling to change their scores), the paper falls below the bar for acceptance.  Hopefully the reviews will help strengthen the paper for a future submission.

**Justification For Why Not Higher Score:**

The paper proposes too many unjustified changes without strongly justifying them in experiments or theory.  There were theoretical proofs, but the reviewers didn't really seem to buy them.  While the paper seems ok.  It just doesn't seem strong enough for acceptance to a top tier conference.

**Justification For Why Not Lower Score:**

NA

---

### Decision · Program_Chairs · 2024-01-16

Reject